# Invariant Convolutional Layers for Time Series

## Abstract

Machine learning for time series has recently garnered considerable attention. Indeed, automatically extracting meaningful representations from large and complex time series data is becoming imperative for several real-world applications. Neural architectures tailored to time series are often built upon sequential modules, such as convolutional, commonly employed in text or vision. Unfortunately, the potential of standard layers in capturing invariant properties of time series remains relatively underexplored. For instance, convolutional layers often fail to capture underlying patterns in time series inputs that encompass strong deformations, such as linear trends. However, invariances to some deformations may be critical for solving complex time series tasks, such as classification, while guaranteeing good generalization properties. To address these challenges, we mathematically formulate and technically design efficient *invariant convolutions* for specific group actions applicable to the case of time series. We construct these convolutions by considering two sets of deformations commonly observed in time series, including (i) *offset shift and scaling* and (ii) *linear trend and scaling*. We further combine the proposed invariant convolutions with standard (or variant) convolutions in a single embedding layer of an example architecture, the so-called INVCONVNET method, and showcase the layer capacity to capture complex invariant time series properties. Finally, INVCONVNET is experimentally proven to achieve superior performance against common baselines in relevant time series tasks, including classification and anomaly detection.

## 1 Introduction

Time series data lies at the core of important real-world applications, spanning multiple scientific and engineering fields, such as energy, transportation, health monitoring and others. Recently, there has been a growing interest in applying machine learning to time series data, with the aim of facilitating several tasks, including prediction, classification, and clustering. However, time series, contrary to other well-structured categories of data, such as images and text, may include variables of different modalities (e. g., weather data consisting of temperature and wind speed measurements) as well as significant noise levels and distribution shifts.

Machine learning for time series has gradually moved from classical statistical methods, e. g., autoregressive models for forecasting (Box et al., 2015) and dictionary-based approaches for classification (Middlehurst et al., 2019), to neural networks. Notable time series neural architectures are built upon recurrent layers, convolutional layers and more recently transformers. Recurrent neural networks (RNNs), originally proposed for sequences and exploited in text-related tasks, suffer from gradient-vanishing problems, especially when modeling long-range dependencies. Attention-based modules, incorporated in transformer architectures, offer a computationally efficient alternative to recurrent architectures, lately achieving state-of-the-art performance in forecasting (Liu et al., 2023; Zhang & Yan, 2022). Nevertheless, it has been shown experimentally that attention modules suffer from optimization issues, e. g., training instability under noisy inputs, which is prominent for the data scarcity of time series. On the other hand, convolutional neural networks (CNNs) have been traditionally applied to extract features from various data sources, including time series. Convolutional layers are also often incorporated in recurrent and attention-based models as initial feature extractors, enhancing the expressiveness of representations for a given task. Indeed, CNNs have been proven consistently successful in tasks including time series classification and clustering (Is-

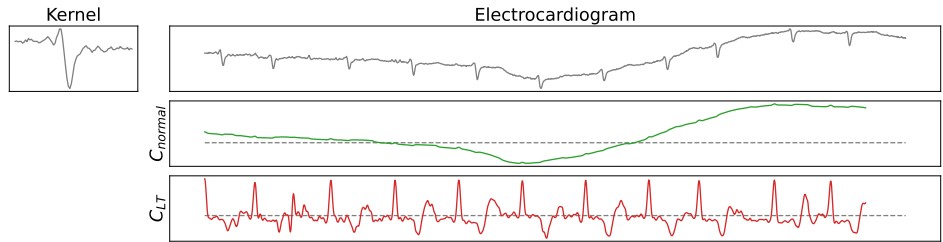

Figure 1: **Top right:** Segment of a electrocardiogram (ECG) from the MIT-BIH dataset Goldberger et al. (2000); Moody & Mark (2001). **Top left:** The convolutional kernel is the first heartbeat. **Middle:** With the normal convolution, individual heartbeats are not identifiable as they are blurred by the trend. **Bottom:** With linear trend invariant convolution, all heartbeat occurrences are identifiable as they are positively correlated with the kernel, minimizing deformations induced by the trend.

mail Fawaz et al., 2019; Tonekaboni et al., 2021), while offering interpretability by visualizing feature's influence through kernel weights.

To address the overfitting issues of standard neural architectures, e. g., transformers (Liu et al., 2020), when dealing with raw time series, significant efforts are focused on the introduction of general-purpose pre-training tasks (Zhang et al., 2024) that incorporate structure in the learning process to overcome data scarcity. For instance, self-supervised contrastive learning for time series, considers transformations of the input to introduce augmentations, and treat them as invariant, via customizable losses (Franceschi et al., 2019; Eldele et al., 2021). Yet, the selection of views within the time series collection to contrast, as well as the types of transformations to consider (e. g., scaling, shifting) are often arguable within the research community (Yue et al., 2022). Notably, soft invariances in place of hard ones, are shown to be experimentally more suitable for capturing complex time series' properties (Lee et al., 2023). However, we need to emphasize here that contrastive learning constitutes an implicit way of introducing invariances into learning rather than via an explicit architectural design choice.

Studying the design of neural layers to be invariant or equivariant to action groups is a notable field in deep learning (Gens & Domingos, 2014; Kondor & Trivedi, 2018). For instance, translation in CNNs and permutation in graph neural networks (GNNs) are two extensively explored properties in network design, among others (Bietti & Mairal, 2019; Horie et al., 2020). Relevant applications range across sets, images, point clouds and graphs (Zaheer et al., 2017; Keriven & Peyré, 2019). Nevertheless, incorporating invariances inside neural layers has not been mathematically formulated and experimentally tested particularly for the case of time series. Indeed, time series representations that are invariant to specific deformations could be advantageous. For instance, in Figure 1, we showcase how a standard convolution with a fixed kernel, selected to match a specific segment (heartbeat) of an ECG series, fails to capture the underlying patterns. On the contrary, a convolution that locally removes the linear trend identifies all underlying heartbeats.

In this work, we aim to mathematically formulate and design convolutional layers that are invariant to certain group actions, specifically tailored to time series data. The main contributions of our study and the proposed layers are summarized as follows:

- We provide theoretical definitions of time series deformations in terms of group actions and introduce the notion of invariance under these actions in Section 3. Based on our theoretical framework, we design a locally invariant embedding to some deformations, e. g., scaling, offset shift, and linear trend, extracted via convolutions. We also propose a strategy that restricts, by learning, the convolutional layer to the relevant invariances for a given task.

- We experimentally test the performance of the proposed layers, by incorporating them in the so-called INVCONVNET model, for time series classification and anomaly detection in Section 4. We perform comparisons with several state-of-the-art baselines, including CNN-based variants of different design choices, e. g., depth, kernel sizes. Furthermore, we provide extensive ablations on the contribution of the different parts of the proposed model, including variant and invariant ones, to synthetic and real-world datasets under different deformations.

## 2    RELATED WORK

**Deep Learning for Time Series.** Dominant deep learning frameworks for time series leverage multi-layer perceptrons (MLPs) (Oreshkin et al., 2019), convolutional networks (CNNs) (Bai et al., 2018) and recurrent ones (RNNs) (Salinas et al., 2020), as well as transformer-based networks (i. e., built upon the attention mechanism) (Wen et al., 2022). Convolutional kernels have traditionally dominated feature extraction in time series, from shapelets (Ye & Keogh, 2011) to the recently successful ROCKET (Dempster et al., 2020), that exploits several random kernels. Additionally, convolutional layers of different kernel sizes, often stacked in deep architectures (Ismail Fawaz et al., 2019), with increased receptive fields, such as INCEPTIONTIME (Ismail Fawaz et al., 2020) and RESNET (Wang et al., 2017), are prominent for time series classification. Similarly, TIMES-NET (Wu et al., 2022) model capitalizes on convolutional layers to capture variations of multiple periodicities of 2D transformed multivariate time series, to solve multiple time series tasks. Beyond standard CNNs for time series, T-WaveNet (Minhao et al., 2021) is a tree-structured wavelet neural network that decomposes the input signal into various frequency subbands with similar energies based on the dominant frequency range. Another recent hierarchical CNN-based model for time series forecasting, SCINET (Liu et al., 2022), repeatedly downsamples and convolves the input to enable information sharing at several resolutions. Furthermore, leveraging the success of the attention mechanism in text, transformer-based architectures have lately proven successful in capturing temporal interactions between multivariate time series inputs, mainly in time series forecasting (Zhou et al., 2021; 2022; Woo et al., 2022; Liu et al., 2021). For instance, AUTOFORMER (Wu et al., 2021) combines decomposition modules with an auto-correlation in place of self-attention, while CROSS-FORMER (Zhang & Yan, 2022) capitalizes on 2D vector array embeddings that preserve temporal and channel information, followed by temporal and channel-wise cross-attention modules. Finally, several recent works, evaluate forecasting architectures also on the anomaly detection (Xu, 2021), by reformulating the task to point-wise reconstruction, with reconstruction error being the anomaly criterion. To overcome the sensitivity of transformers in overfitting, recent simple MLP-based architectures have showcased superior performance in forecasting, e. g., TSMIXER (Chen et al., 2023), FRETS (Yi et al., 2024), with several studies challenging the overall effectiveness of such complex architectures for time series (Zhang et al., 2022; Zeng et al., 2023).

**Invariances for Time Series Modeling.** Besides the supervised setting described above, multiple modern methods emphasize self-supervised techniques for extracting representations from time series data, before the downstream task. Inspired by other domains (He et al., 2022), masked autoencoders trained to reconstruct missing time points across samples form a common choice for unsupervised pre-training, with primary applications in forecasting (Nie et al., 2022; Dong et al., 2024). In the same category of unsupervised approaches, but focusing on incorporating knowledge about similarities between representations, lies contrastive self-supervised learning. Specifically, invariance is often achieved by applying transformations (e. g., scaling, shifting, or noise injection) to the input and training the model to recognize that these altered versions should map to the same underlying representation (Chen et al., 2020). Typically, CNNs constitute basic blocks for various time series augmentation-based contrastive frameworks, such as TS-TCC (Eldele et al., 2021) and TIMECLR (Yang et al., 2022). Except for building augmented views solely with transformations, samples are also contrasted with some sampled subseries (Franceschi et al., 2019), with adjacent segments (Tonekaboni et al., 2021), or a combination of both along with transformations (Yue et al., 2022). Unlike deep learning, invariances in time series data have long been a central focus in classical time series data mining approaches (Esling & Agon, 2012). For instance, local time warping invariance (Ding et al., 2008) can be tackled by dynamic time warping (DTW). Additionally, amplitude and offset invariances are accomplished by Z-normalizing the data (Paparrizos et al., 2020). Finally, LT-normalized distance (Germain et al., 2024) extends Z-normalized distance by being invariant to linear trend in addition to amplitude and offset shifts, facilitating similarity search and motif set discovery applications.

## 3    METHOD

In this section, we provide all the essential formulations for the introduction of our *invariant convolutional layers* and propose their incorporation as embedding layers in architectures, i. e., the so-called INVCONVNET for time series modeling.

## 3.1 Invariant embedding for time series

We, next, develop a mathematical framework to create an embedding invariant to a predefined set of deformations. Essentially, the embedding is expected to map a geometrical object or any of its deformed versions to the same representative. In addition, we present some important properties that should verify the embedding and tailor the proposed framework to the case of time series.

**Deformations and group action.** From a geometrical viewpoint, the notion of invariance depends on the representation of deformations and the definition of the action of a deformation on a geometrical object. A classical approach consists of representing a deformation as an element of a group and its action by a group action:

**Definition 3.1** (Group action). *A group* G *with neutral* e *acts on the left on a set* M*, if there exists a map* $a :$ G $\times$ M $\mapsto$ M *that verifies:*

$$1) \; a(e, m) = m, \quad \forall m \in \mathsf{M}$$

$$2) \; a(g, a(h, m)) = a(gh, m), \quad \forall (g, h) \in \mathsf{G}^2, \forall m \in \mathsf{M}.$$

To simplify notation, the left action of $g \in$ G on $m \in$ M is denoted $g \cdot m$. For a group G that acts on the left on a set M, the orbit of $m \in$ M is the set of all its deformed versions $[m] = \{g \cdot m \mid g \in \mathsf{G}\}$. The set of independent orbits, denoted M/G, is called quotient space, and if this set is reduced to a singleton, the action of G on M is said transitive, and it verifies that for any $m \in$ M its orbit is the whole set: $[m] =$ M.

**A group action for time series.** Leveraging measure theory, we model the set of time series by the Hilbert space $\mathsf{L}^2(\mathsf{I}, \mathbb{R}^D, \mu)$ of functions defined on the closed interval $\mathsf{I} \subset \mathbb{R}$ taking value in $\mathbb{R}^D$ and square-integrable for the Borel measure $\mu$. The inner product on $\mathsf{L}^2(\mathsf{I}, \mathbb{R}^D, \mu)$ is defined as:

$$\langle f, g \rangle_{\mathsf{L}} = \int_{\mathsf{I}} \langle f(t), g(t) \rangle d\mu(t) \tag{1}$$

where $\langle ., . \rangle$ is the dot product on $\mathbb{R}^D$. Let H be a finite dimensional vector subspace of $\mathsf{L}^2(\mathsf{I}, \mathbb{R}^D, \mu)$, we model the group of deformations as the set $\mathbb{R}_+^* \ltimes$ H with the composition rule $(\lambda_2, h_2) \times (\lambda_1, h_1) = (\lambda_2 \lambda_1, h_2 + \lambda_2 h_1)$. Finally, we model the group action by the application:

$$((\lambda, h), f) \in (\mathbb{R}_+^* \ltimes \mathsf{H}) \times \mathsf{L}^2(\mathsf{I}, \mathbb{R}^D, \mu) \mapsto \lambda f + h \in \mathsf{L}^2(\mathsf{I}, \mathbb{R}^D, \mu) \tag{2}$$

This is a general group action that is not transitive as H is a finite-dimensional vector subspace of $\mathsf{L}^2(\mathsf{I}, \mathbb{R}^D, \mu)$. By convention, we refer to $\mathbb{R}_+^* \ltimes$ H as the set of rigid deformations. The customization of the group action depends on the choice of basis for the subspace H. For instance, the Z-normalization (Paparrizos et al., 2020) is an invariant offset shift which corresponds to the subspace of deformations $\{h : \mathsf{I} \mapsto c \mid c \in \mathbb{R}^D\}$ with the basis $\{h_i : \mathsf{I} \mapsto e_i/\sqrt{length(\mathsf{I})} \mid i \in [1, \ldots, D]\}$ where $(e_i)_{i \in [1, \ldots, D]}$ is the orthonormal basis of $\mathbb{R}^D$.

**Invariant embedding.** An embedding invariant to a group action is expected to map any element of an orbit to the same representative, and it is defined as follows:

**Definition 3.2** (Invariant & orbit-injective embedding). *An embedding map* $L :$ M $\mapsto$ N *is said to be* G*-invariant, if for any* $(g, m) \in$ G $\times$ M*,* $L(g \cdot m) = L(m)$. *Additionally,* L *is said to be orbit-injective if the application* $\tilde{L} : [m] \in$ M/G $\mapsto L(m) \in$ N *is injective.*

Note that an invariant embedding is meaningful in the case of a non-transitive group action. In addition, if the embedding is orbit-injective, each orbit has a distinct representative.

For now, we focus on the action of the finite-dimensional subspace H of a Hilbert space M by the usual vector addition: $(h, m) \in$ H $\times$ M $\mapsto m + h \in$ M. The following proposition exhibits a H-invariant embedding that is also orbit-injective.

**Proposition 1.** *Let* $P_{\mathsf{H}}$ *be the orthogonal projector on* H*, and* $I_d$ *be the identity map on* M*, the embedding,* $L = I_d - P_{\mathsf{H}}$ *(the projector on* $\mathsf{H}^{\perp}$*) is* H*-invariant and orbit-injective.*

*Proof.* See Appendix A.1. □

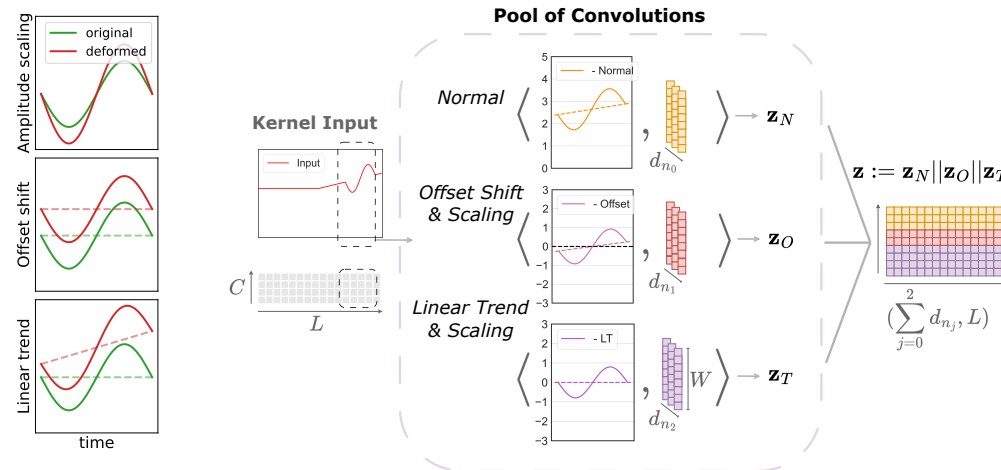

Figure 2: **Left:** Different types of deformations applied to an example series, including amplitude scaling, offset shift, and linear trend. **Right:** Visualization of the different kernel types employed on an input signal inside the proposed invariant convolutional layer, including normal filters (in yellow), filters invariant to offset shift (in red) and filters invariant to linear trend (in purple). The produced embedding $z$ is the result of concatenations of the different representations.

**Remark 1.** *If* $(h_i)_{i \in [\![1,N]\!]}$ *is an orthonormal basis of the finite dimensional vector subspace* H, *then the orthogonal projector on* H *as an explicit formulation* $P_{\mathsf{H}} : m \in \mathsf{M} \mapsto \sum_{i=1}^{N} \langle m, h_i \rangle_{\mathsf{L}} h_i \in \mathsf{H}$.

Invariance to amplitude scaling can easily be incorporated in an embedding defined by the previous proposition:

**Proposition 2.** *Let* $L : \mathsf{M} \mapsto \mathsf{M}$ *be the* H-*invariant and orbit-injective embedding map induced by the orthogonal projector on* H *as defined in proposition 1. The embedding map:*

$$\hat{L} : \ m \in \mathsf{M} \mapsto \left\{ \begin{array}{ll} L(m)/\|L(m)\|_{\mathsf{M}} & \textit{if } m \in \mathsf{M} \backslash \mathsf{H} \\ 0_{\mathsf{M}} & \textit{else} \end{array} \right. \tag{3}$$

*is* $(\mathbb{R}_+^* \ltimes \mathsf{H})$-*invariant and orbit-injective.*

*Proof.* $(\mathbb{R}_+^* \ltimes \mathsf{H})$-invariance is due to the linearity and H-invariance of $L$, and the orbit-injectivity in induced by the linearity and orbit-injectivity of $L$. □

**An example: the univariate Z-normalization.** We are looking for an embedding invariant to amplitude scale and offset shift in the case of univariate discrete time series. The set of time series is modeled by $\mathsf{L}^2([0, l], \mathbb{R}, \mu)$ where $l \in \mathbb{N}^*$, $\mu = \sum_{i=1}^{l} \delta_i$ and $\delta_i$ is the dirac measure at $i$. The set offset shifts is the subspace generated by the unit norm function $e : \ t \in [0, l] \mapsto 1/\sqrt{l} \in \mathbb{R}$. According to Proposition 2 the invariant embedding of a non-constant function $f$ is the function: $(f - \langle f, e \rangle_L e)/\|f - \langle f, e \rangle_L e\|_L$ which leads to $(f(i) - \mu_f)/(\sqrt{l}\sigma_f)$ where $\mu_f = l^{-1} \sum_{i=1}^{l} f(i)$ and $\sigma_f^2 = l^{-1} \sum_{i=1}^{l} (f(i) - \mu_f)^2$.

### 3.2 INVARIANT CONVOLUTION

CNNs have been successful in many applications related to time series, essentially becoming a key building block of the latest deep neural networks. Their success comes from their ability to capture local information in long time series. However, CNNs remain sensitive to some deformations like amplitude scaling or offset shifts (Mallat, 2016). In this section, we propose a novel convolution that is invariant to rigid deformations at a local scale while remaining computationally efficient.

**The formalism.** Let $\mathsf{L}^2_{loc}(\mathbb{R}, \mathbb{R}^D, \mu)$ be the set of signals, we assume that the signal in square integrable on any compact of $\mathbb{R}$. Let $\mathsf{L}^2(\mathsf{I}, \mathbb{R}^D, \mu)$ be the set of kernels where $\mathsf{I} \subset \mathbb{R}$ is a closed interval. The classical convolution layer, named 1D-CNN, between a signal $f$ and a kernel $w$ is the signal:

$$f * w : u \in \mathbb{R} \mapsto \int_\mathsf{I} \langle f(u+t), w(t) \rangle d\mu(t) \in \mathbb{R} \tag{4}$$

Let assume a group of rigid deformations $\mathsf{G}$ acting on $\mathsf{L}^2(\mathsf{I}, \mathbb{R}^D, \mu)$, and $\hat{L}$ the $\mathsf{G}$-invariant embedding map defined by Proposition 2. For any $u \in \mathbb{R}$, we can define the operator $K_u^\mathsf{G}$ that maps the restriction of any signal $f$ on the closed interval $u + \mathsf{I}$ to its $\mathsf{G}$-invariant representative:

$$K_u^\mathsf{G} : f \in \mathsf{L}^2_{loc}(\mathbb{R}, \mathbb{R}^D, \mu) \mapsto \hat{L}\left(t \in \mathsf{I} \mapsto f(t+u)\right) \in \mathsf{L}^2(\mathsf{I}, \mathbb{R}^D, \mu) \tag{5}$$

Leveraging these operators we define the $\mathsf{G}$-invariant convolution between a signal $f$ and a kernel $w$ as the signal:

$$f *^\mathsf{G} w : u \in \mathbb{R} \mapsto \int_\mathsf{I} \langle (K_u^\mathsf{G} f)(t), w(t) \rangle d\mu(t) \in \mathbb{R} \tag{6}$$

**Fast computation.** For a group of rigid deformations $(\mathbb{R}_+^* \ltimes \mathsf{H})$ with $(h_i)_{i \in [\![1,N]\!]}$ a basis of $\mathsf{H}$, thanks to Remark 1, the inner product between the invariant representation of $f \in \mathsf{L}^2(\mathsf{I}, \mathbb{R}^D, \mu)$ and $w$ can be decomposed as follows: $\langle \hat{L}(f), w \rangle_\mathsf{L} = (\langle f, w \rangle_\mathsf{L} - \sum_{i=1}^N \langle f, h_i \rangle_\mathsf{L} \langle w, h_i \rangle_\mathsf{L})/\|L(f)\|_\mathsf{L}$. Assuming discrete signals, the computation of $\langle \hat{L}(f), w \rangle_\mathsf{L}$ requires the computation of $2N+2$ dot products. However, convolving a batch of $B$ signals of length $L$ with the kernel, the number of inner products to compute drops from $BL(2N+2)$ to $BL(N+2)+N$ as the inner products between the kernel and the basis are shared across signals and subsequences. It leads to the time complexity $\mathcal{O}(BLNCW)$ where $C$ is the number of channels, $W$ is the kernel size and assuming that $N << L$. Invariant convolutions do not consider small-size kernels (2 or 3 timestamps) but rather large kernels (30 or more). The traditional approach to convolution is not tractable in such a context. Instead, we leverage the Fast Fourier transform (FFT) (Mathieu et al., 2013), which changes the time complexity to $\mathcal{O}(BNCL \log(L))$. The computational time is identical for any window size, as the computation with the FFT does not depend on the kernel size. In the experimental results, we indeed show that our proposed invariant convolutions benefit from fast computation.

**Pool of convolutions.** The choice of invariances is often related to the application (Yue et al., 2022), and setting the invariances by hand requires a good understanding of the nature of the signals. In the absence of such knowledge, one possible strategy consists of decomposing the space of deformations $\mathsf{H} = \bigoplus_{i=1}^K \mathsf{H}_i$ in the direct sum of subspaces such that the cumulative sums, $\emptyset \subset \mathsf{H}_1 \subset \mathsf{H}_1 + \mathsf{H}_2 \subset \ldots \subset \mathsf{H}$, represent sets of deformations of increasing order of complexity. We represent a layer as the concatenation of $n_j$ $(\bigoplus_{i=1}^j \mathsf{H}_i)$-invariant convolutions for $j \in (1, \ldots, K)$ and $n_0$ standard convolutions. In the experiments that follow, we show that the decomposition of invariances in subspaces enables the learning of application-specific invariances while dropping irrelevant ones. Note that other strategies, are possible, notably the use of attention mechanisms, and this is let for future work.

In the following experiments, the most complex deformations considered are the linear trends: $\mathsf{H} = \{t \in \mathsf{I} \mapsto at + b \in \mathbb{R}^D \mid (a,b) \in \mathbb{R}^D \times \mathbb{R}^D\}$. This vector space can be decomposed in the action of the offset shifts $\mathsf{H}_1 = \{t \in \mathsf{I} \mapsto b \mid b \in \mathbb{R}^D\}$ and the purely linear deformations $\mathsf{H}_2 = \{t \in \mathsf{I} \mapsto at \mid a \in \mathbb{R}^D\}$. Therefore, we consider layers composed of standard convolutions, convolutions invariant to offset shift and amplitude scaling, and finally, convolutions invariant to linear trend and amplitude scaling.

Figure 2 (Left) shows the basic deformations considered, i. e., amplitude scaling, offset shift, and linear trend, whereas Figure 2 (Right) provides a visualization of the proposed invariant convolutional layer, which incorporates three different kernel types, to capture the invariances. Additional details about the construction of the embedding layers for different time series tasks built upon invariant convolutions can be found in the Appendix A.2. For any task, the embedding module is identical, and it is a single layer (of depth 1) of invariant convolutions with kernels of potentially different sizes capturing features of different scales. The embedding module is followed by a task-specific

Table 1: Classification performance for the considered time series datasets. Average accuracy (%) is mentioned for all combinations of models and datasets. Accuracy is averaged for all datasets in *UEA* repository. Higher is better, best methods in **bold**, second best underlined.

| Datasets | INVCONVNET (ours) | TIMESNET (2022) | PATCHTST (2022) | CROSSFORMER (2022) | TSLANET (2024) | DLINEAR (2023) | INCEPTION (2020) | RESNET (2017) | CNN (2018) | ROCKET (2020) |
|---|---|---|---|---|---|---|---|---|---|---|
| *UEA* (26 datasets) | **71.81 ± 0.80** | 66.87 ± 1.72 | 66.18 ± 1.26 | 66.37 ± 1.35 | 68.70 ± 1.19 | 61.51 ± 1.05 | 62.86 ± 1.96 | 67.37 ± 1.59 | 65.67 ± 1.64 | 71.29 ± 0.90 |
| *UCIHAR* | **96.63 ± 0.49** | 91.66 ± 0.62 | 85.74 ± 0.50 | 93.43 ± 0.56 | 94.71 ± 0.66 | 57.47 ± 0.73 | 95.26 ± 0.55 | 96.04 ± 0.48 | 95.78 ± 0.20 | 92.06 ± 0.15 |
| *Sleep-EDF* | 84.95 ± 0.39 | 74.64 ± 0.73 | 78.53 ± 0.28 | 79.82 ± 0.89 | 84.98 ± 0.43 | 36.15 ± 0.21 | 84.06 ± 0.39 | **85.62 ± 0.13** | 82.41 ± 0.56 | 83.88 ± 0.09 |
| *Epilepsy* | **98.43 ± 0.04** | 97.62 ± 0.20 | 98.01 ± 0.05 | 98.23 ± 0.12 | 98.23 ± 0.05 | 82.26 ± 0.06 | 97.65 ± 0.20 | 98.16 ± 0.04 | 97.61 ± 0.28 | 98.38 ± 0.02 |

final module. For classification, the final module is an average-pooling layer followed by a linear layer. For tasks trained by reconstruction, the final module refines the embedding output with a standard CNN to capture multiple scale dependencies. The decoder that follows takes as input the refined embedding as well as the coefficients from the signal decomposition on the invariant basis.

# 4 EXPERIMENTAL EVALUATION

We next perform an extended experimental evaluation for the proposed INVCONVNET model, built upon the previously defined invariant convolutions. We focus on setups that can benefit from invariances, such as those concerning scale offset and linear trends. Intuitively, those naturally arise in time series classification and anomaly detection, presented in Section 4.1 and Section 4.2.

## 4.1 CLASSIFICATION

To evaluate the proposed invariant convolutional layers, we first focus on the long-standing problem of classification, including univariate and multivariate time series across different applications.

### 4.1.1 EXPERIMENTAL SETTING

**Datasets.** We evaluate INVCONVNET on the 26 multivariate *UEA* data repository (Bagnall et al., 2018), coming with a standard train/test split. We train our INVCONVNET on the raw inputs, which are padded for the case of non-equal length series, while for the baselines, each variable is normalized independently using Z-normalization. We also consider 3 additional datasets, the *UCIHAR* (Anguita et al., 2013) human activity recognition (HAR) dataset, the *Sleep-EDF* dataset (Goldberger et al., 2000) for sleep stage classification of EEG signals, and finally the *Epilepsy* dataset (Andrzejak et al., 2001), which is an epileptic seizure recognition dataset. For all three datasets, we follow the exact same preprocessing with (Eldele et al., 2021), deriving train/validation/test sets of $60 : 20 : 20$ ratio. For the synthetic experiment involving added invariances, we selected the four largest datasets from the *UCR* repository (Dau et al., 2019). Finally, we perform a transfer learning experiment using 4 different source and target domains of the *Fault-Diagnosis* dataset (Lessmeier et al., 2016). Additional details about the datasets can be found in the Appendix A.3.

**Baselines.** We select nine state-of-the-art models for time series classification from the relevant literature. The CNN-based model TIMESNET (Wu et al., 2022), the transformer-based methods PATCHTST (Nie et al., 2022) and CROSSFORMER (Zhang & Yan, 2022) and the MLP-based architecture DLINEAR (Zeng et al., 2023) are derived from the Time-Series-Library (Wang et al., 2024). Additionally, we perform comparisons with the CNN-based backbone of the self-supervised method TSLANET (Eldele et al., 2024), which replaces attention with a Fourier-based spectral block and interactive 1D CNNs for capturing temporal variations. We also consider 3 powerful CNN architectures capturing different depths and receptive fields, including INCEPTION (Ismail Fawaz et al., 2020) and RESNET (Wang et al., 2017), which are constructed using standard convolutional layers with varying filter sizes organized into successive blocks with residual connections, and the simpler CNN (Ismail Fawaz et al., 2018) that employs several stacked convolutional layers of a fixed kernel size with varying hidden dimensions. As a powerful machine learning baseline for time series, we incorporate ROCKET (Dempster et al., 2020). Finally, for the proposed INVCONVNET, we select for each dataset among the 3 introduced variants, i. e., the standard one with one convolutional layer, the inception-like, and the multi-scale one, as presented in Appendix A.2.

Table 2: Study on the effect of the different kernel types of INVCONVNET in classification accuracy, including solely normal (or variant) kernels (-N), or offset-invariant kernels (-O), or trend-invariant kernels (-T).

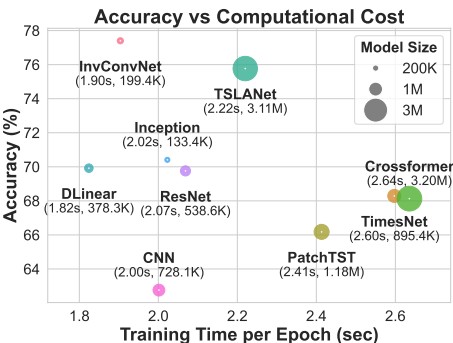

Figure 3: Cost comparisons on *Heartbeat*.

| Datasets | INVCONVNET (ours) | INVCONVNET-N - *normal* - | INVCONVNET-O - *offset* - | INVCONVNET-T - *trend* - |
|---|---|---|---|---|
| *UEA* (26 datasets) | **71.81 ± 0.80** | 68.04 ± 1.76 | 67.70 ± 1.21 | 66.61 ± 2.58 |
| *UCIHAR* | **96.63 ± 0.49** | 96.13 ± 0.34 | 95.75 ± 0.51 | 95.43 ± 0.13 |
| *Sleep-EDF* | **84.95 ± 0.39** | 84.70 ± 0.39 | 83.73 ± 0.13 | 83.14 ± 0.13 |
| *Epilepsy* | **98.43 ± 0.04** | 98.09 ± 0.13 | 98.26 ± 0.04 | 98.25 ± 0.09 |

### 4.1.2 RESULTS

**Performance comparisons.** We present in Table 1 the classification performance of the proposed INVCONVNET along with the eight considered deep learning-based baselines and one classical convolutional-based machine learning algorithm, ROCKET. All algorithms, are evaluated on the 26 *UEA* datasets and 3 additional data sources, i. e., *UCIHAR*, *Sleep-EDF* and *Epilepsy*. All models are trained and tested for 3 runs with random seeds and we reported the average accuracy along with its standard deviation. We observe that on *UEA* repository, INVCONVNET has the best average test classification accuracy, followed closely by ROCKET, and both algorithms outperform the other considered deep learning approaches. Full classification results per dataset on *UEA* can be found in the Appendix A.5. Indeed, ROCKET employs thousands of randomly initialized convolutional kernels to efficiently extract diverse features from the time series inputs that are then fed into a ridge regression classifier. This high number of random kernels explains the robustness of ROCKET on the smaller *UEA* classification datasets. On the three remaining datasets, INVCONVNET also shows a superior performance in terms of accuracy, proving again the advantage of invariant CNN-based approaches in classification. For *Sleep-EDF*, RESNET is slightly better than the proposed INVCONVNET, which can be attributed to the depth of the method in capturing complex dependencies between the time series inputs. Finally, transformer-based methods are significantly outcompeted by CNN-based ones, and the MLP-based DLINEAR scores the worst, failing to capture the class-dependent temporal dynamics in this task. Additionally, the proposed INVCONVNET built upon a single layer of convolutional kernels shows significant advantage in terms of time and memory cost (time per epoch, number of model parameters), as presented in Figure 3 for the *Heartbeat* dataset from *UEA*.

**Ablation study.** Table 2 contains the results of an ablation study that compares the proposed INVCONVNET with its standalone main components on the same classification experiments. More specifically, the standard INVCONVNET combines three types of convolutions: the normal ones (similar to conventional convolutions), those invariant to offset shift, and those invariant to linear trend. In this experiment, we compare INVCONVNET with other configurations of the network that conserve the same architecture but consider only one type of convolution: normal for INVCONVNET-N, offset shift invariant for INVCONVNET-O, and linear trend invariant for INVCONVNET-T. For INVCONVNET, the number of kernels per convolution type is identical, and in all cases, the total number of kernels remains the same. As depicted in Table 2 INVCONVNET shows the highest performance on all datasets. It is followed by INVCONVNET-N that only considers conventional convolution, which proves the significance of including invariant convolutions within the pools of convolutions for classification. Additionally, the solely invariant configurations -B, -C, surpass most baselines on all datasets, and even CNN-based ones (such as INCEPTION, CNN), despite being shallow compared to the latter deep architectures.

**Synthetic experiment.** We also conduct a synthetic experiment on the 4 larger datasets from the *UCR* archive (Dau et al., 2019), following the same setup as the ablation study. We deformed each dataset according to 5 scenarios: (i) *no deformations*, (ii) *the addition of random offset*, sampled from uniform distribution between specific ranges, as well as (iii) *the addition of random trend* with slope and intercept values sampled again with uniform probability, (iv) *combination of added ran-*

Table 3: Robustness study of INVCONVNET and its standalone declinations, i. e., -N (normal), -O (offset) or -T (trend), on the 4 larger *UCR* datasets under 4 scenarios of synthetic deformations: (i) *no deformations*, (ii) *the addition of random offset (off.)* (iii) *the addition of random linear trend (LT)*, (iv) *combination of added random offset and linear trend (off., LT)* and (v) *combination of added random offset and smooth random walk (off., RW).* Higher is better, best methods in **bold**.

| Datasets | | INVCONVNET (ours) | INVCONVNET-N - normal - | INVCONVNET-O - offset - | INVCONVNET-T - trend - |
|---|---|---|---|---|---|
| *HandOutlines* | *Normalized* | 74.80 | **80.60** | 72.70 | 69.82 |
| | *+ off.* | 71.20 (-4.8%) | 68.70 (-14.8%) | **71.60** (-1.5%) | 70.18 (+0.5%) |
| | *+ LT* | **74.00** (-1.1%) | 72.90 (-9.6%) | 71.70 (-1.4%) | 70.18 (+0.5%) |
| | *+ off., LT* | 71.70 (-4.1%) | 70.70 (-12.3%) | 70.20 (-3.4%) | **71.71** (+2.7%) |
| | *+ off., RW* | 65.68 (-12.2%) | 61.89 (-23.2%) | **67.83** (-6.7%) | 66.22 (-5.2%) |
| *UWaveGestureLibraryAll* | *Normalized* | 83.40 | **85.00** | 75.60 | 71.50 |
| | *+ off.* | **81.10** (-2.8%) | 79.00 (-7.0%) | 70.10 (-7.3%) | 69.40 (-2.9%) |
| | *+ LT* | 82.40 (-1.2%) | **83.80** (-1.4%) | 74.20 (-1.9%) | 71.30 (-0.3%) |
| | *+ off., LT* | **81.40** (-2.4%) | 79.00 (-7.0%) | 70.00 (-7.4%) | 68.10 (-4.8%) |
| | *+ off., RW* | **74.04** (-11.2%) | 68.90 (-18.9%) | 68.76 (-9.0%) | 68.79 (-3.8%) |
| *StarLightCurves* | *Normalized* | 96.00 | **96.50** | 95.40 | 93.40 |
| | *+ off.* | **97.30** (+1.4%) | 94.70 (-1.9%) | 94.10 (-1.4%) | 90.00 (-3.6%) |
| | *+ LT* | **97.30** (+1.4%) | 95.20 (-1.3%) | 94.50 (-0.9%) | 91.90 (-1.6%) |
| | *+ off., LT* | **97.50** (+1.6%) | 92.30 (-4.4%) | 95.10 (-0.3%) | 90.10 (-3.5%) |
| | *+ off., RW* | **93.21** (-2.9%) | 81.61 (-15.4%) | 90.98 (-4.6%) | 90.80 (-2.8%) |
| *MixedShapesRegularTrain* | *Normalized* | **95.30** | 94.50 | 87.40 | 92.60 |
| | *+ off.* | **94.20** (-1.2%) | 87.30 (-7.6%) | 90.60 (+3.7%) | 91.92 (-0.7%) |
| | *+ LT* | **94.60** (-0.7%) | 92.40 (-2.2%) | 89.10 (+1.9%) | 92.10 (-0.5%) |
| | *+ off., LT* | 91.80 (-3.7%) | 86.60 (-8.4%) | 86.20 (-1.4%) | **92.40** (-0.2%) |
| | *+ off., RW* | **90.89** (-4.6%) | 72.62 (-23.2%) | 88.16 (+0.9%) | 87.46 (-5.6%) |

Table 4: Classification Accuracy (%) on a Transfer Learning experiment on *Fault-Diagnosis* for the supervised INVCONVNET and INVCONVNET-N (normal) and two self-supervised methods.

| Methods | $A \rightarrow B$ | $A \rightarrow C$ | $A \rightarrow D$ | $B \rightarrow A$ | $B \rightarrow C$ | $B \rightarrow D$ | $C \rightarrow A$ | $C \rightarrow B$ | $C \rightarrow D$ | $D \rightarrow A$ | $D \rightarrow B$ | $D \rightarrow C$ | Avg. Acc. (%) |
|---|---|---|---|---|---|---|---|---|---|---|---|---|---|
| TS-TCC *(FT)* | 55.33 ± 1.44 | 52.52 ± 4.55 | 62.13 ± 1.39 | 48.05 ± 3.32 | 71.50 ± 1.83 | 100.0 ± 0.0 | 40.76 ± 2.22 | 98.25 ± 1.22 | 99.34 ± 0.50 | 46.98 ± 0.65 | 100.0 ± 0.0 | 74.28 ± 2.77 | 70.76 ± 1.66 |
| TS2VEC *(FT)* | 54.11 ± 1.46 | 54.07 ± 1.91 | 52.54 ± 1.89 | 55.06 ± 0.17 | 88.72 ± 0.47 | 100.0 ± 0.0 | 57.81 ± 2.18 | 78.30 ± 3.80 | 78.41 ± 4.39 | 60.37 ± 1.95 | 99.97 ± 0.02 | 86.82 ± 0.54 | 72.18 ± 1.57 |
| INVCONVNET *(Sup.)* | 55.90 ± 0.42 | **55.93 ± 0.34** | 53.41 ± 0.14 | **85.10 ± 0.63** | 78.54 ± 0.17 | 99.05 ± 0.08 | **70.75 ± 1.32** | 85.04 ± 0.13 | 85.12 ± 0.15 | **70.91 ± 0.73** | **100.0 ± 0.0** | 78.49 ± 0.38 | **76.52 ± 0.37** |
| INVCONVNET-N *(Sup.)* | **60.55 ± 0.88** | 55.50 ± 1.82 | 53.50 ± 0.85 | 60.26 ± 2.01 | 77.30 ± 0.60 | 93.50 ± 0.90 | 64.93 ± 0.48 | 84.87 ± 0.23 | 84.46 ± 0.51 | 59.98 ± 0.98 | 99.96 ± 0.0 | 77.14 ± 0.14 | 72.66 ± 0.78 |

*dom offset and trend* and (v) *combination of random offset shift and smooth random walk. For the last deformation, the added synthetic trend is a random walk generated from a Gaussian distribution and smoothed by a rolling mean.* In Table 3, we display the classification accuracy for each pair of dataset/scenario and networks with different convolution pool configurations: INVCONVNET-N only normal, INVCONVNET-O only offset shift invariant, INVCONVNET-T only linear trend invariant, INVCONVNET all 3 types. We also mention inside parenthesis the percentage increase (in blue) or decrease (in red) of accuracy with respect to the one achieved on the plain data (scenario (i)) for each configuration. Interestingly, we observe that for the synthetic deformations, our model INVCONVNET maintains relatively high performance and outcompetes, in most cases, the INVCONVNET-N configuration that solely considers conventional convolutions.

**Transfer Learning Experiment.** We evaluate the generalization properties of the proposed INVCONVNET model on a transfer learning experiment compared to the popular self-supervised methods TS-TCC (Eldele et al., 2021) and TS2VEC (Yue et al., 2022). Additionally, we show results for the INVCONVNET-N configuration built upon only normal (not invariant) filters. More specifically, all models are trained and tested on different source and target domains, referring to the different A, B, C, and D sub-datasets of *Fault-Diagnosis*. The self-supervised methods are pre-trained and fine-tuned (FT) on each source domain dataset by leveraging contrastive learning. Results in terms of accuracy are demonstrated in Table 4. Interestingly, the supervised INVCONVNET model built upon invariant convolutions significantly outperforms both unsupervised methods and its normal configuration (-N), offering accuracy improvements of at least 4% with a small variance in performance.

## 4.2 ANOMALY DETECTION

Anomaly detection identifies unusual patterns in time series. Reconstruction-based models learn to reconstruct the input, and the error is the anomaly criterion based on a chosen threshold.

### 4.2.1 EXPERIMENTAL SETTING

**Datasets.** We evaluate the proposed model in unsupervised anomaly detection for time series, aiming to identify anomalous time points. To achieve this, we choose five common time series anomaly

Table 5: Anomaly Detection results for several datasets. Performance mentioned in terms of the F1-score (%) for all combinations of models and datasets. Higher is better, best methods in **bold**, second best underlined.

| Datasets | INVCONVNET (ours) | TIMESNET (2022) | PATCHTST (2022) | TSLANET (2024) | ETSFORMER (2022) | FEDFORMER (2022) | LIGHTTS (2022) | DLINEAR (2023) | AUTOFORMER (2021) | PYRAFORMER (2021) | INFORMER (2021) | REFORMER (2020) |
|---|---|---|---|---|---|---|---|---|---|---|---|---|
| *SMD* | 84.05 ± 0.16 | **84.61 ± 0.56** | 84.15 ± 0.48 | 84.33 ± 0.17 | 79.69 ± 0.69 | 71.11 ± 0.02 | 83.04 ± 0.49 | 83.56 ± 0.14 | 71.16 ± 0.02 | 71.36 ± 0.01 | 71.17 ± 0.03 | 71.22 ± 0.01 |
| *MSL* | 80.68 ± 0.01 | 80.33 ± 0.79 | 78.67 ± 0.04 | 74.65 ± 0.78 | 75.98 ± 0.54 | **82.06 ± 0.14** | 80.39 ± 0.06 | 81.92 ± 0.01 | **82.08 ± 0.04** | 81.00 ± 0.08 | 82.02 ± 0.11 | 81.52 ± 0.08 |
| *SMAP* | 68.29 ± 0.07 | 69.18 ± 0.21 | 68.84 ± 0.01 | **80.26 ± 0.05** | 67.45 ± 0.74 | 68.71 ± 0.01 | 67.47 ± 0.02 | 67.32 ± 0.01 | 75.28 ± 1.64 | 67.76 ± 0.13 | 68.74 ± 0.12 | 73.30 ± 0.15 |
| *SWaT* | **92.82 ± 0.19** | 92.71 ± 0.04 | 88.38 ± 1.11 | 91.65 ± 0.27 | 92.67 ± 0.06 | 79.18 ± 0.01 | 92.75 ± 0.01 | 92.66 ± 0.01 | 79.18 ± 0.01 | 80.91 ± 0.38 | 79.75 ± 0.74 | 79.17 ± 0.01 |
| *PSM* | 96.34 ± 0.01 | **96.85 ± 0.27** | 96.12 ± 0.01 | 96.20 ± 0.03 | 95.23 ± 0.03 | 89.44 ± 0.88 | 95.50 ± 0.02 | 96.66 ± 0.01 | 88.25 ± 0.01 | 93.66 ± 0.13 | 90.55 ± 0.05 | 90.74 ± 0.09 |
| **Avg. F1 (%)** | 84.44 ± 0.09 | 84.74 ± 0.37 | 83.23 ± 0.33 | **85.42 ± 0.26** | 82.20 ± 0.41 | 78.10 ± 0.21 | 83.83 ± 0.12 | 84.42 ± 0.04 | 79.19 ± 0.65 | 78.94 ± 0.15 | 78.45 ± 0.21 | 79.19 ± 0.07 |

detection datasets; the *SMD* dataset (Su et al., 2019), the *MSL* and *SMAP* datasets (Hundman et al., 2018), *SWaT* (Mathur & Tippenhauer, 2016) and *PSM* (Abdulaal et al., 2021), describing problems of server monitoring, environmental exploration and monitoring, and critical infrastructure systems. We follow standard preprocessing methods to extract successive non-overlapping sub-sequences and split them into train/validation/test sets with a $70 : 10 : 20$ ratio (Xu, 2021; Wu et al., 2022).

**Baselines.** Since we focus on reconstruction, we mainly deploy models commonly used in time series regression. We select ten state-of-the-art models for time series from which most can be found in the Time-Series-Library (Wang et al., 2024), including the CNN-based TIMESNET (Wu et al., 2022), the transformer-based methods PATCHTST (Nie et al., 2022), ETSFORMER (Woo et al., 2022), FEDFORMER (Zhou et al., 2022), AUTOFORMER (Wu et al., 2021), PYRAFORMER (Liu et al., 2021), INFORMER (Zhou et al., 2021), REFORMER (Kitaev et al., 2020), and the MLP-based architecture LIGHTTS (Zhang et al., 2022) and DLINEAR (Zeng et al., 2023), as well as the recent CNN-based TSLANET (Eldele et al., 2024). For the proposed INVCONVNET, we use the same embedding blocks as those used in classification. To perform reconstruction of the input, we use linear layers for the temporal dimension and then for the channel dimension, as described in the Appendix A.2. We utilize the same decoder also for the standard CNN-based architectures, i. e., INCEPTION, RESNET and CNN.

### 4.2.2 RESULTS

Table 5 displays anomaly detection scores in terms of F1-score (%) for the proposed model and the baselines on the 5 anomaly detection datasets. We observe that the proposed INVCONVNET model is the best-performing method on *SWaT* dataset, whereas overall, is the third in terms of average performance for all datasets, slightly surpassed by the CNN-based TIMESNET model. This can be attributed to the refinement of the CNN blocks in TIMESNET to account for multiple periodicities, allowing finer granularities for reconstruction. Similarly, TSLANET is the best-performing model in terms of average F1, and it leverages Fourier blocks prior to CNNs modules to capture features of short and long dependencies simultaneously. Other types of architectures like the purely MLP-based framework DLINEAR, and the transformer-based FEDFORMER also show competitive performances for several datasets. Interestingly, INVCONVNET achieves strong performance across all datasets while using a single layer of invariant convolutions with kernels of different sizes. This demonstrates that leveraging invariances across multiple scales can be highly effective even in shallow architectures for reconstruction-based anomaly detection. Finally, we provide in the Appendix A.5 performance comparisons between INVCONVNET and the simple CNN-based embedding modules of INCEPTION, RESNET, CNN, stressing the importance of including invariances and their related coefficients for the reconstruction task.

## 5 CONCLUSION

In this paper, we mathematically formulate the crucial modeling aspect of invariances when dealing with time series data. We leverage this formulation to design invariant convolutions that we carefully combined with conventional convolutions in a single network layer, and we propose an example architecture, i. e., the INVCONVNET model. Experimental results show that our proposed invariant embedding modules benefit from competitive performance in different tasks, while remaining computationally attractive and shallow. Incorporating our modules in general-purpose architectures that leverage layers of different types, e. g., attention, and unsupervised pre-training, e. g., via masking, enabling additional applications, remains on our agenda for future work.

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

# A    APPENDIX

## A.1    INVARIANT EMBEDDING

Let M be a Hilbert space and H a finite dimensional vector subspace of M. We focus on the action H on M by the usual vector addition: $(h, m) \in H \times M \mapsto m + h \in M$. The following proposition exhibits a H-invariant embedding that is also orbit-injective.

**Proposition 3.** *Let $P_H$ be the orthogonal projector on* H*, and $I_d$ be the identity map on* M*, the embedding, $L = I_d - P_H$ (the projector on* $H^\perp$*) is* H*-invariant and orbit-injective.*

*Proof.* **Existence of** $L$: As H is a finite dimension vector space, it is a closed and convex subset of the Hilbert space M; the orthogonal projector on H, denoted $P_H$, exists. Therefore, $L : m \in M \mapsto m - P_H(m) \in H$ is well defined.

H**-invariance of** $L$: Since H is closed, $M = H \oplus H^\perp$, and for any $x \in M$, we decompose $m = m_H + f_{H\perp}$. Thus, for any $m \in M$, and $h \in H$:

$$
\begin{aligned}
L(m + h) &= m + h - P_H(m + h) \\
&= m + h - P_H(m_{H\perp} + m_H + h) \\
&= m + h - (m_H + h) \quad \textbf{(projector on a closed vectorial subspace)} \\
&= m - m_H \\
&= L(m)
\end{aligned}
$$

which proves the H-invariance of $L$.

**Orbit-injectivity of** $L$**:** For any $m \in M$, its orbits corresponds to:

$$
\begin{aligned}
[m] &= \{m + h \mid h \in H\} \\
&= \{L(m) + h' \mid h \in H, \ h' = P_H(m) + h \in M\} \\
&= L(m) + H
\end{aligned}
$$

Therefore, for any $([m], [m']) \in M/H \times M/H$, such that $[m] \cap [m'] = \emptyset$ implies that $L(m) \neq L(m')$ proving the orbit-injectivity of $L$. $\qquad\square$

## A.2    INVCONVNET: ARCHITECTURAL DETAILS

### A.2.1    INVARIANT EMBEDDING MODULES

After mathematically formulating the characteristics of an invariant convolutional layer, which is built upon a standard (or variant) kernel, a kernel invariant to offset shift and scaling, and a kernel invariant to linear trend and scaling, we provide additional details for the design of the employed embedding modules. We present visualizations of the embedding modules used for classification and anomaly detection (i. e., reconstruction) in Figure 4.

**Standard Module (Single-Layer):** The simplest embedding module is a single invariant convolutional layer for a specific kernel size $W$ and hidden dimensions $d_{n_0}$ for the standard convolutional part (in yellow), $d_{n_1}$ for the convolutional part invariant to offset shift and scaling (in red), and $d_{n_2}$ for the convolutional part invariant to linear trend and scaling (in purple).

**Inception-like Module (Single-Layer):** We also study inception-like design by employing several kernel sizes, but without stacking the layers in increasing depths. The depth of the employed module remains equal to one. As shown in Figure 4 (Left), in an inception-like embedding module, we consider several kernel sizes, e. g., $W_1, W_2, W_3$, that are applied in parallel to the input series, while leveraging the three parts of the proposed pool of convolutions (including standard and invariant ones). The produced representation for the different kernel sizes, i. e., $\mathbf{z_1}, \mathbf{z_2}, \mathbf{z_3}$ are concatenated in the channel dimension producing embeddings of size $(3 * \sum_{j=0}^{2} d_{n_j}, L)$ for 3 selected kernel sizes. The distinct kernel sizes as well as the hidden dimension for each part in the pool of convolutions, are hyperparameters that we need to tune, as in every CNN-based architecture.

**Multi-Scale Module (Multi-Layer):** Additionally, we examine the capacity of a multi-scale embedding module built upon invariant convolutions as presented in Figure 4 (Right), particularly for the reconstruction task. Here the employed depth is equal to two. At the first level, an inception-like

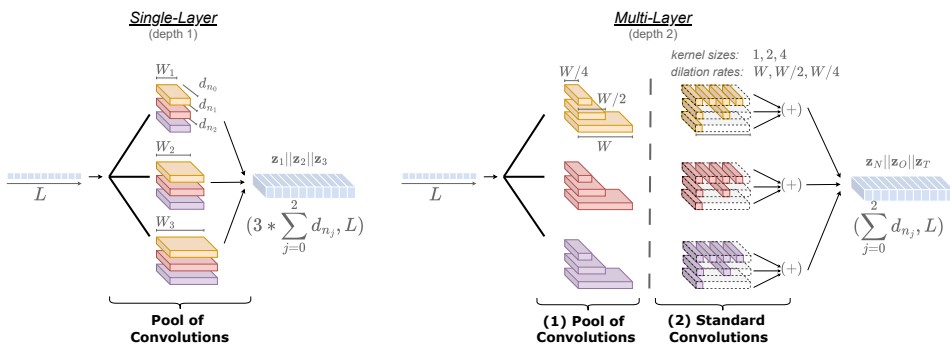

Figure 4: **Left:** Single-layer embedding module, i.e., of depth 1, used in standard INVCONVNET (for 1 chosen kernel width $W$) and in inception-like INVCONVNET (for several chosen kernel widths, e.g., 3 visualized in the figure). **Right:** Multi-layer embedding module, i.e., of depth 2, used in multi-scale INVCONVNET (for several chosen kernel widths $W$). In the second layer, for each kernel size, we utilize multiple kernels whose total number sums up to the larger kernel size, achieving multi-scale views.

layer is employed, with kernel sizes selected to be powers of two (deriving the maximum exponent from the logarithm of half the series length and setting the minimum to four). The kernels, as described above, are applied in parallel, and the produced representation for the different kernel sizes is concatenated in the channel dimension. At the second level, a standard convolutional layer is applied. We similarly employ several kernel sizes of size $\max(W_i)/W_i$ matching the picked kernel sizes in the first layer $W_i$ for $i \in \{1, \ldots, K\}$, where $K$ the number of kernels with distinct sizes. For each distinct kernel size in this second layer, a dilation factor is set as $r_i = W_i$, thus equal to the kernel size of the previous layer. This design enables capturing representations at different scales while employing a shallow and computationally light architecture, that still benefits from invariant convolutions. Experimentally, this module shows performance improvements for the anomaly detection task, where reconstruction can benefit from capturing dependencies at different granularities.

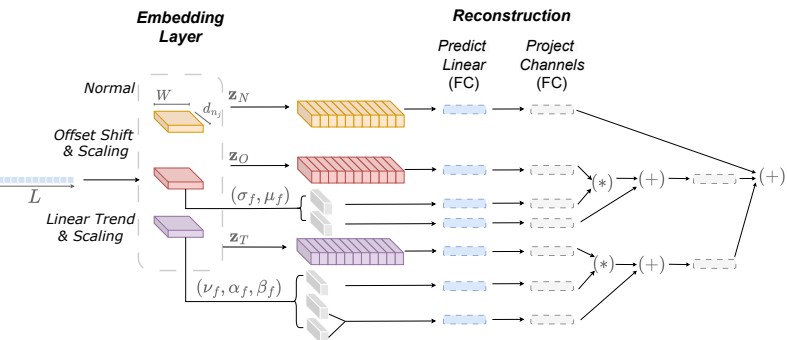

Figure 5: Visualization of the architecture used in terms of reconstruction that leverages the output of an embedding module built upon a pool of convolutions (including variant and invariant ones). The representation of the embedding layer $\mathbf{z}_N, \mathbf{z}_O, \mathbf{z}_T$ are passed from 2 fully connected linear layers (FC), from which the first operates on the temporal dimension and the second on the channels. The coefficients of each invariant operation are similarly projected with linear layers and combined with the representation (with addition or multiplication) to produce the output.

### A.2.2 TASK-SPECIFIC MODULES

The embeddings derived from the modules described above are further processed by standard layers to produce the task-specific output.

**I. Classification.** The embedding is passed by a Global Average Pooling (GAP) layer, applied to the channel dimension, that averages over the time dimension to reduce the temporal features to a single value per channel. The result of the GAP layer is followed by a linear layer that produces the final class probabilities.

**II. Anomaly Detection (Reconstruction).** The multi-layer embedding module presented above is used to capture dependencies at various scales. For the reconstruction of the input, the embedding is followed by linear layers applied first on the temporal and then on the channel dimensions. The coefficients of the invariant kernels, i. e., $(\sigma_f, \mu_f)$ for the offset-invariant part and $(\nu_f, \alpha_f, \beta_f)$ for the trend-invariant part, are passed through linear layers to be mapped to the original time and channel dimensions and are then combined with each part (one variant and two invariant) of the embedding via addition and multiplication. More specifically, $\sigma_f$ and $\nu_f$ refer to the norm of invariant embeddings of the signal, $\mu_f$ refers to the means, $\alpha_f$ and $\beta_f$ refers to the coefficients of the linear trend. Combining the coefficients with the linearly projected embeddings adapts the level of the series to the original one, enabling enhanced temporal resolution. Figure 5 presents details about the reconstruction module and its relation with the embedding module.

## A.3 DATASETS DETAILS

We focus our experimental evaluation on several real-world time series datasets, including univariate and multivariate inputs, with significant applications in healthcare and medical diagnosis, wearable technology, audio processing, and transportation, among others.

**Classification Datasets.** Details about the 26 multivariate derived from *UEA* data repository (Bagnall et al., 2018), that are employed in terms of this study in the classification experiment are provided in Table 6. More specifically, for each dataset we mention the number of channels, the length of the multivariate series, as well as the number of classes and the number of instances in the predefined train and test sets.

Table 6: Details of UEA datasets used for classification.

| Dataset | #Train | #Test | #Channels | Length | #Classes |
|---|---|---|---|---|---|
| *ArticularyWordRecognition* | 275 | 300 | 9 | 144 | 25 |
| *AtrialFibrillation* | 15 | 15 | 2 | 640 | 3 |
| *BasicMotions* | 40 | 40 | 6 | 100 | 4 |
| *Cricket* | 108 | 72 | 6 | 1197 | 12 |
| *Epilepsy* | 137 | 138 | 3 | 206 | 4 |
| *EthanolConcentration* | 261 | 263 | 3 | 1751 | 4 |
| *FaceDetection* | 5890 | 3524 | 144 | 62 | 2 |
| *FingerMovements* | 316 | 100 | 28 | 50 | 2 |
| *HandMovementDirection* | 320 | 147 | 10 | 400 | 4 |
| *Handwriting* | 150 | 850 | 3 | 152 | 26 |
| *Heartbeat* | 204 | 205 | 61 | 405 | 2 |
| *InsectWingbeat* | 30000 | 20000 | 200 | 78 | 10 |
| *JapaneseVowels* | 270 | 370 | 12 | 29 | 9 |
| *Libras* | 180 | 180 | 24 | 51 | 6 |
| *LSST* | 2459 | 2466 | 6 | 36 | 14 |
| *MotorImagery* | 278 | 100 | 64 | 3000 | 2 |
| *NATOPS* | 180 | 180 | 24 | 51 | 6 |
| *PEMS-SF* | 267 | 173 | 963 | 144 | 7 |
| *PenDigits* | 7494 | 3498 | 2 | 8 | 10 |
| *PhonemeSpectra* | 3315 | 3353 | 11 | 217 | 39 |
| *RacketSports* | 151 | 152 | 6 | 30 | 4 |
| *SelfRegulationSCP1* | 268 | 293 | 6 | 896 | 2 |
| *SelfRegulationSCP2* | 200 | 180 | 7 | 1152 | 2 |
| *SpokenArabicDigits* | 6599 | 2199 | 13 | 93 | 10 |
| *StandWalkJump* | 12 | 15 | 4 | 2500 | 3 |
| *UWaveGestureLibrary* | 120 | 320 | 3 | 315 | 8 |

Table 7 contains the same details for the additional datasets used for classification, i. e., the *UCIHAR* (Anguita et al., 2013) dataset, the *Sleep-EDF* dataset (Goldberger et al., 2000), and the *Epilepsy* dataset (Andrzejak et al., 2001). More specifically, *UCIHAR* data were collected by 30 volunteers performing various activities, including laying, standing, sitting, walking, walking downstairs, and walking upstairs. Volunteers' records were captured by a waist smartphone, including distinct measurements connected to acceleration and velocity signals. The *Sleep-EDF* dataset from the PhysioBank database consists of PolySomnoGraphic sleep recordings containing EEG, among other

Table 7: Details of rest datasets used for classification.

| Dataset | #Train | #Test | #Channels | Length | #Classes |
|---------|--------|-------|-----------|--------|----------|
| *UCIHAR* | 7352 | 2947 | 9 | 128 | 6 |
| *Sleep-EDF* | 25612 | 8910 | 1 | 3000 | 5 |
| *Epilepsy* | 9200 | 2300 | 1 | 178 | 2 |
| *Fault-Diagnosis* | 8184 | 2728 | 1 | 5120 | 3 |

Table 8: Details of datasets used for anomaly detection.

| Dataset | #Train | #Val | #Test | #Channels | Length |
|---------|--------|------|-------|-----------|--------|
| *SMD* | 566724 | 141681 | 708420 | 38 | 100 |
| *MSL* | 44653 | 11664 | 73729 | 55 | 100 |
| *SMAP* | 108146 | 27037 | 427617 | 25 | 100 |
| *SWaT* | 396000 | 99000 | 449919 | 51 | 100 |
| *PSM* | 105984 | 26497 | 87841 | 25 | 100 |

measurements. We consider only the EEG signals following previous studies (Eldele et al., 2021) and performed sleep stage classification, including awake, rapid eye movement, and non-rapid eye movements. Finally, the *Epilepsy* dataset consists of EEG brain activity measurements for epileptic seizure classification. Following the preprocessing of (Eldele et al., 2021), we perform binary classification after merging classes referring to non-epileptic seizure.

We also conduct a synthetic experiment on the 4 larger datasets from *UCR* repository (Dau et al., 2019). Since many *UCR* datasets are already preprocessed using Z-normalization to achieve zero mean and unit variance, they are not ideal for demonstrating the impact of our invariant layers on classification performance. This is the primary reason for conducting a synthetic experiment on these datasets rather than a conventional one with the whole repository. For the transfer learning experiment, we utilized the *Fault-Diagnosis* dataset (Lessmeier et al., 2016), as preprocessed in (Eldele et al., 2021), which comprises of measurements under 4 different working conditions, perceived as different domains, and are assigned to 3 classes, including a healthy and two fault classes.

**Anomaly Detection Datasets.** Furthermore, we present in Table 8 the five employed anomaly detection datasets after preprocessing them on non-overlapping subsequences of length 100, also showing the number of channels and the size of the train, validation, and test splits. The *SMD* dataset (Su et al., 2019) consists of data related to server machines collected at an internet company, while the *MSL* and *SMAP* (Hundman et al., 2018) datasets comprise of telemetry data from spacecraft monitoring systems. The *SWaT* (Mathur & Tippenhauer, 2016) dataset is a collection of sensor data from the operations of a critical infrastructure system. Finally, the *PSM* (Abdulaal et al., 2021) dataset contains measurements from application server nodes on an internet website.

**Data Splits and Preprocessing.** As mentioned already in the main paper, for the proposed method, we do not normalize the data using Z-normalization for *UEA* and the rest 4 datasets used in classification, while the datasets from *UCR* are used for the synthetic experiment are derived normalized by the data source. On the contrary, all data are normalized for classification and the baselines, as well as for anomaly detection and all considered models (including the proposed INVCONVNET). For the *UEA* datasets, we do validation on the whole training set since the test sets are, in several cases, quite large, and thus, a small subset of the train set picked for validation can be a misleading indicator of performance. For the rest classification datasets, we perform a split into train/validation/test sets with a $60 : 20 : 20$ ratio, following (Eldele et al., 2021). Similarly, for the five anomaly detection datasets, we split into train/validation/test sets with a $70 : 10 : 20$ ratio (Xu, 2021).

### A.4 IMPLEMENTATION DETAILS

All experiments presented in this study were conducted on an Nvidia Tesla V100 GPU, with 40 cores and 756 GB of memory. We utilized the Adam optimizer with a learning rate of lr = 0.001 for both classification and unsupervised anomaly detection tasks. We also adopted a linear cosine annealing learning rate scheduler for INVCONVNET in classification. More specifically, the scheduler started the warmup phase with a learning rate equal to 0.001, linearly increasing the learning rate over the first 10 epochs to 0.01. After the warmup, it gradually reduced the learning rate using a cosine annealing schedule, down to 0.0001 by the end of training. For anomaly detection and the rest methods, we utilized a learning rate scheduler of 0.5 decrease rate per epoch. To have better esti-

mates for the generalization performance of all models and, most importantly, our proposed shallow modules, we performed 3 runs with random seeds for all considered datasets and tasks. Additional details for each task and the hyperparameters of the models are given below.

**- Classification Task:** We trained the models for 100 epochs for all *UEA* datasets, the 4 *UCR* datasets and the *Fault-Diagnosis* dataset. We performed early stopping during training, after 20 epochs of no improvement in the validation accuracy for all models and kept the configuration of weights that correspond to the best validation accuracy during training. The standard cross entropy loss was optimized during training for classification. For the INVCONVNET model, we considered the inception-like embedding module of Figure 4 (Left) for all datasets of *UEA* except for *Epilepsy*, *EthanolConcentration*, *Heartbeat* that we selected the standard embedding module of single kernel size. Finally, for *FingerMovements*, *Handwriting*, *Libras* and *SelfRegulationSCP1* datasets, we employed the multi-scale embedding layer of Figure 4 (Right). The type of the embedding layer, as well as the hyperparameters for the convolutional layers, e.g., kernel size and hidden dimension, were selected through random search and the best performance on the validation set. For the rest of the classification datasets, i.e., the *UCIHAR*, the *Sleep-EDF* and the *Epilepsy* datasets, we trained all models for 300 epochs with 20 epochs patience and considered the inception-like embedding layer, since it was performing better on the validation set.

**- Unsupervised Anomaly Detection Task:** We trained the models for 10 epochs and stopped training if no improvements had been made in terms of validation loss for 3 epochs, saving the best model weights on the validation set. We optimized the models using the mean squared error (MSE) between the real input sequences and the reconstructed ones. For all five anomaly detection datasets, we used the multi-scale embedding layer of Figure 4 (Right), followed by the reconstruction module built upon linear layers in Figure 5.

**- Hyperparameter Selection:** We next provide more information about the selection of the kernel sizes and hidden dimensions for the different embedding modules tested in terms of INVCONVNET. For the standard pool of convolutions with one specific kernel size $W$, we chose the kernel size as the minimum value between the value $50$ and half of the length of the time series. For the inception-like embedding module, we selected several kernel sizes, such as $51, 75, 101$, and $125$, or factors of those values for which the length of the series is proportional. Finally, for the first layer (pool of convolutions) of the multi-scale module, we computed the kernel sizes as powers of two, starting from $16$ up to a maximum of $128$, based on the logarithmic scaling of half the series length. For all modules, we tested hidden dimensions sizes for the pool of convolutions in $\{32, 64, 128, 256\}$ doing a split that enabled almost equal contribution for the three parts, i.e., normal, invariant to offset shift and scaling, and invariant to linear trend and scaling. For instance for total hidden size equal to $32$ the different parts had $(12, 10, 10)$ hidden dimensions respectively, for $64$ the split became $(24, 20, 20)$ and so on.

For the common CNN-based baselines INCEPTION, RESNET, CNN, we tuned the number of convolutional layers, the kernel sizes, and the hidden size of each layer. We followed a random search for a value between $2$ and $6$ for the number of blocks and $\{32, 64, 128, 256\}$ for the hidden dimensions, whereas for the kernel sizes, we used those proposed in the relevant papers (Ismail Fawaz et al., 2020; Wang et al., 2017; Ismail Fawaz et al., 2018). All baselines' implementations are derived from the Time-Series-Library (Wang et al., 2024), with the configurations mentioned in the respective papers, and the main code resources for performing the different tasks, e.g., classification and anomaly detection were adopted. We also used ROCKET (Dempster et al., 2020) from sktime Library (Löning et al., 2019), with 3000 random convolutional kernels. Finally, for the transfer learning classification experiment, the self-supervised contrastive TS-TCC and TS2VEC methods were trained with their default parameters for classification as proposed in the respective papers (Eldele et al., 2021; Yue et al., 2022), for 50 epochs for each phase of pre-training and fine-tuning.

## A.5 Additional Results

### A.5.1 Synthetic Experiment - Visualization of Feature Maps

In Figure 6, we provide visualizations of the feature maps produced for the example INVCONVNET architecture and its INVCONVNET-N (normal) declination for particular cases of the synthetic experiment presented in Table 3. Specifically, we consider one out of the 4 larger *UCR* datasets, namely the *MixedShapesRegularTrain* dataset.

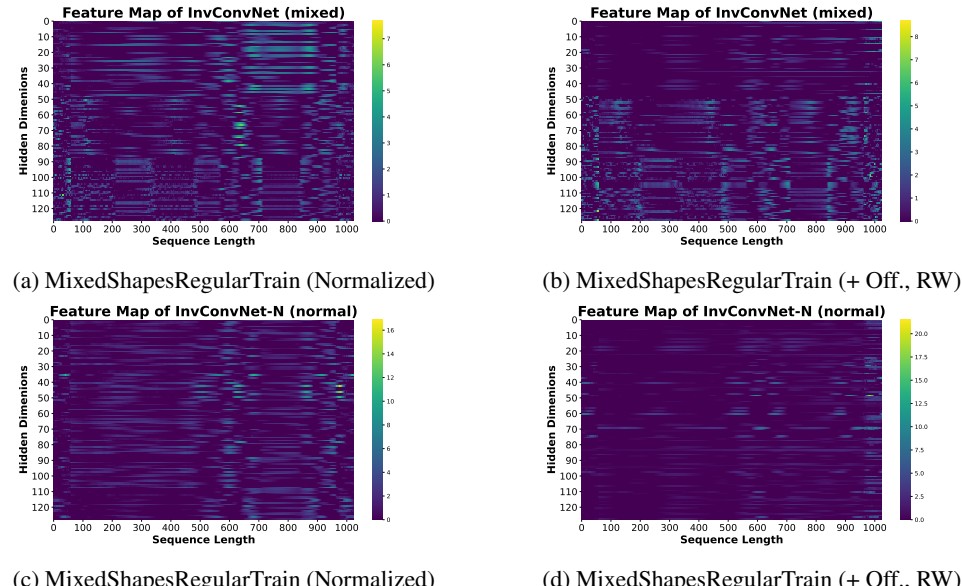

(a) MixedShapesRegularTrain (Normalized)      (b) MixedShapesRegularTrain (+ Off., RW)

(c) MixedShapesRegularTrain (Normalized)      (d) MixedShapesRegularTrain (+ Off., RW)

Figure 6: Comparison of the feature maps produced in the synthetic classification experiment (Table 3) on samples of *MixedShapesRegularTrain* dataset by the proposed convolutional layer before average-pooling as generated by: (a) & (b) the proposed INVCONVNET (mixed) built upon different types of invariances including (i) *no invariance*, (ii) *filters invariant to offset shift and amplitude scaling* and (iii) *filters invariant to linear trend and amplitude scaling*, (c) & (d) INVCONVNET-N (normal) built upon normal convolutional filters (non-invariant). We consider as inputs the plain *MixedShapesRegularTrain* dataset, which is already normalized (in (a) and (c)), or the synthetically deformed data with an added random offset and smooth random walk (in (b) and (d)).

For the considered dataset, we select one sample that is correctly classified for the first class for both INVCONVNET (mixed) and INVCONVNET-N (normal). INVCONVNET (mixed) is built upon three types of convolutions (normal, invariant to offset shift, and invariant to linear trend), and the standalone declination INVCONVNET-N (normal) is solely built upon normal convolutional filters. We recall here that each model configuration (i. e., INVCONVNET, INVCONVNET-N) has been separately trained and tested for the plain data and each case of synthetically deformed data.

**Extraction of Feature Maps.** After passing the input series through the convolutional layer of each model and the activation function (i. e., ReLU(.)), we extract the feature map for each filter (or each hidden dimension) corresponding to the largest considered kernel size. We recall that for the synthetic experiment on the 4 larger *UCR* datasets, we leveraged the inception-like embedding module of Figure 4 (Left), built upon 4 different kernel sizes with each having an equal total number of filters (or hidden dimensions equal to 128). Please note that averaging over all outputs produced by the layer for the several distinct kernel sizes produces multi-scale representations that produce similar (in terms of activated regions) but smoother maps (in terms of intensity values). The construction of the layer differs depending on the number of filters considered for each type of convolution, and for the first two model configurations becomes as follows: INVCONVNET has $(48, 40, 40)$ filters respectively for each type of convolution (i. e., standard or non-invariant, invariant to offset shift and invariant to linear trend) and the baseline case INVCONVNET-N has $(128, 0, 0)$ filters, thereby only considers 128 standard filters. The resulting feature maps, which are derived by the activated outputs for the largest kernel size, are essentially 2D representations with dimensions equal to the series length $L$ and the 128 hidden dimensions.

We provide in Figure 6 the activated feature maps for a single sample for *MixedShapesRegularTrain* dataset, extracted for the two considered model configurations (INVCONVNET (mixed), INVCONVNET-N (normal) as heatmaps. Color in the heatmap plots corresponds to the magnitude of the activation at a specific location of the series for each hidden dimension, with a lighter color (i. e., yellow) representing higher activations. Plots on the left column (sub-figures 6a, 6c) correspond to a

plain sample from the dataset without any added deformation (that is already Z-normalized) whereas plots on the right column (sub-figures 6b, 6d) correspond to the same sample with added random offset and smooth random walk generated trend (+ *Off., RW*).

**Deformation-Specific Activation Maps for Different Types of Filters.** The example INVCON-VNET model, which combines three types of convolutions (standard, offset-invariant, and trend-invariant), exhibits three distinct regions in its feature maps, each corresponding to one of these convolution types. This is evident for the values of the activated feature maps for different hidden dimensions, as shown in the first row of Figure 6 (i.e., the sub-figures 6a, 6b). Specifically, the activated regions for the first 48 dimensions display a different morphology compared to the next 40 dimensions, which, in turn, differ from the final 40 dimensions. This highlights the unique contribution of each convolution type (or invariance) to the model's learned representations inside the so-called ***pool of convolutions*** (introduced in Figure 2). On the contrary, INVCONVNET-N, which is built solely upon normal convolutions, exhibits a distinct morphology of activated features that appears more homogeneous across the various hidden dimensions (i.e., the sub-figures 6c, 6d).

More specifically, we observe that the normal filters (i.e., first 48 hidden dimensions) of INVCON-VNET model are more activated for a raw (normalized) sample in Figure 6a. When random offset and a smooth random walk trend are added to the sample, the normal filters are less activated compared to filters invariant to offset shift and linear trend in Figure 6b (hidden dimensions $48 - 88$ and $88 - 128$ respectively). Importantly, activations remain quite similar for invariant filters (last 80 dimensions) between plain and deformed data (as shown in sub-figures 6a and 6b), as the kernels are not locally affected by the deformations.

On the other hand, for the purely normal filters in INVCONVNET-N, the activated feature maps exhibit very high values around specific time points, likely corresponding to abrupt changes or hollowed areas in the time series input (as shown in sub-figure 6c). Notably, when a smooth random walk trend is added as a deformation, the feature maps of normal filters in INVCONVNET-N become smoother. This can be attributed to a localized reduction in the effect of abrupt changes (in sub-figure 6d). Consequently, under the added deformation, INVCONVNET-N seems to struggle to identify meaningful regions in the input signal, with the activated feature maps being dominated and absorbed by the synthetic trend. On the contrary, for this particular added deformation, apart from the less informative normal filters in the activated feature maps, offset-invariant and trend-invariant filters are activated at several local levels of the input time series (as shown in sub-figure 6b).

### A.5.2 CLASSIFICATION

We also provide in Table 9 the full classification results for the 26 considered *UEA* datasets that correspond to the average of 3 runs for each combination of dataset and model. In the same table, we include again the already presented in the main paper, average accuracy for the whole collection of datasets as well as the number where each model scores first in the last row. The *JapaneseVowels* dataset is mentioned as out-of-time ('OOT') for not producing performance results since the experiment did not run within the time limits (12 hours maximum for each dataset). From the full classification results, we observe that the proposed INVCONVNET is, in several cases, slightly outperformed by the classical ROCKET method, but on average, is among the first best-competing models for most datasets, which explains its performance superiority in terms of average accuracy for the whole *UEA*.

In several studies (Wu et al., 2022; Zhou et al., 2023), only a subset of 10 *UEA* datasets is considered, and we also present once again the results for this subset along with total the average accuracy in 10. Similar observations can be made as those for Table 9, with the proposed INVCONVNET scoring the best average accuracy of 73.22%, followed by ROCKET.

### A.5.3 ANOMALY DETECTION

Finally, in Table 11, we perform additional comparisons, in terms of anomaly detection, including the INVCONVNET model and the standard CNN-based variants, namely INCEPTION, RESNET and CNN originally proposed for classification. All models have an identical reconstruction module, with the exception of INVCONVNET, which also includes the signal decomposition coefficients on the invariant basis.

Table 9: Full Classification results for UEA datasets. Accuracy (%) is mentioned for all combinations of models and datasets. Higher is better, best methods in **bold**, second best underlined.

| Dataset | INVCONVNET | TIMESNET | PATCHTST | CROSSFORMER | TSLANET | DLINEAR | INCEPTION | RESNET | CNN | ROCKET |
|---|---|---|---|---|---|---|---|---|---|---|
| *ArticularyWordRecognition* | 99.00 | 97.78 | 97.67 | 98.22 | 98.22 | 96.67 | 84.56 | 98.44 | 97.89 | **99.44** |
| *AtrialFibrillation* | 37.78 | 28.89 | **42.22** | 28.89 | 24.44 | 35.56 | 28.89 | 24.44 | 33.33 | 6.67 |
| *BasicMotions* | **100.00** | 95.00 | 70.83 | 91.67 | **100.00** | 81.67 | 87.50 | **100.00** | **100.00** | **100.00** |
| *Cricket* | 98.61 | 93.06 | 94.44 | 92.59 | 97.69 | 91.20 | 87.96 | 98.15 | 98.61 | **100.00** |
| *Epilepsy* | 95.89 | 89.61 | 97.34 | 87.44 | 96.86 | 51.45 | 92.27 | 94.44 | 92.27 | **98.55** |
| *EthanolConcentration* | 25.98 | 26.24 | 23.32 | **39.67** | 22.18 | 24.97 | 23.57 | 21.93 | 22.81 | 29.40 |
| *FaceDetection* | 64.71 | **67.50** | 64.77 | 65.26 | 56.59 | 62.97 | 63.88 | 54.82 | 52.75 | 59.13 |
| *FingerMovements* | **56.33** | 55.00 | 53.33 | 52.33 | 55.00 | 48.67 | **56.33** | 53.00 | 53.00 | 54.00 |
| *HandMovementDirection* | 40.99 | **64.41** | 47.75 | 57.21 | 45.50 |  | 31.08 | 36.04 | 29.28 | 44.59 |
| *Handwriting* | 53.14 | 28.67 | 26.98 | 26.39 | 48.71 | 18.71 | 17.22 | 37.10 | 36.20 | **56.27** |
| *Heartbeat* | **77.40** | 68.29 | 66.18 | 68.13 | 75.77 | 69.92 | 70.41 | 69.76 | 62.76 | 73.17 |
| *InsectWingbeat* | 'OOT' | 'OOT' | 'OOT' | 'OOT' | 'OOT' | 'OOT' | 'OOT' | 'OOT' | 'OOT' | 'OOT' |
| *JapaneseVowels* | 97.66 | 91.71 | 94.68 | 96.76 | 96.85 | 93.33 | 91.80 | **98.83** | 98.38 | 97.39 |
| *Libras* | 88.70 | 79.07 | 76.11 | 86.30 | 84.81 | 50.19 | 57.04 | **94.07** | 88.70 | 91.11 |
| *LSST* | 55.04 | 12.77 | 48.35 | 11.21 | 10.41 | 31.85 | 35.71 | 8.99 | 9.37 | **60.76** |
| *MotorImagery* | 49.67 | 52.00 | 50.67 | **55.00** | 47.67 | 50.33 | 51.67 | 51.33 | 51.33 | 46.33 |
| *NATOPS* | 95.74 | 93.33 | 75.00 | 87.41 | 94.63 | 90.74 | 90.74 | **96.67** | 95.74 | 87.96 |
| *PEMS-SF* | 80.35 | 78.61 | 81.89 | **84.39** | 79.96 | 80.15 | 75.53 | 79.38 | 74.37 | 80.15 |
| *PenDigits* | 98.78 | 98.48 | 97.52 | 97.12 | 98.12 | 87.32 | 97.75 | 98.70 | **98.81** | 98.08 |
| *PhonemeSpectra* | **29.82** | 14.31 | 12.62 | 12.63 | 26.71 | 6.72 | 21.91 | 28.66 | 27.15 | 27.69 |
| *RacketSports* | 87.72 | 82.68 | 76.75 | 79.82 | 88.16 | 67.98 | 83.77 | **90.57** | 84.43 | 90.35 |
| *SelfRegulationSCP1* | 86.12 | **87.60** | 78.84 | 85.32 | 79.18 | 83.39 | 81.57 | 80.32 | 84.64 | 84.53 |
| *SelfRegulationSCP2* | 54.44 | 48.15 | 45.19 | 47.59 | 53.89 | 45.74 | 53.33 | 48.15 | 48.33 | **54.82** |
| *SpokenArabicDigits* | 99.47 | 98.83 | 97.92 | 98.67 | **99.58** | 95.85 | 98.50 | 99.23 | 98.98 | 99.56 |
| *StandWalkJump* | 28.89 | 33.33 | **51.11** | 24.44 | 48.89 | 33.33 | 26.67 | 40.00 | 31.11 | 48.89 |
| *UWaveGestureLibrary* | 92.92 | 86.35 | 83.02 | 84.69 | 87.71 | 77.92 | 61.77 | 81.35 | 71.56 | **93.33** |
| **Avg. Accuracy (%)** | **71.81** | 66.87 | 66.18 | 66.37 | 68.70 | 61.51 | 62.86 | 67.37 | 65.67 | 71.29 |
| **1st Count** | 4 | 3 | 2 | 3 | 2 | 0 | 1 | 5 | 2 | 8 |

Table 10: Full Classification results for a subset of 10 UEA datasets. Accuracy (%) is mentioned for all combinations of models and datasets. Higher is better, best methods in **bold**, second best underlined.

| Dataset | INVCONVNET | TIMESNET | PATCHTST | CROSSFORMER | TSLANET | DLINEAR | INCEPTION | RESNET | CNN | ROCKET |
|---|---|---|---|---|---|---|---|---|---|---|
| *EthanolConcentration* | 25.98 | 26.24 | 23.32 | **39.67** | 22.18 | 24.97 | 23.57 | 21.93 | 22.81 | 29.40 |
| *FaceDetection* | 64.71 | **67.50** | 64.77 | 65.26 | 56.59 | 62.97 | 63.88 | 54.82 | 52.75 | 59.13 |
| *Handwriting* | 53.14 | 28.67 | 26.98 | 26.39 | 48.71 | 18.71 | 17.22 | 37.10 | 36.20 | **56.27** |
| *Heartbeat* | **77.40** | 68.29 | 66.18 | 68.13 | 75.77 | 69.92 | 70.41 | 69.76 | 62.76 | 73.17 |
| *JapaneseVowels* | 97.66 | 91.71 | 94.68 | 96.76 | 96.85 | 93.33 | 91.80 | **98.83** | 98.38 | 97.39 |
| *PEMS-SF* | 80.35 | 78.61 | 81.89 | **84.39** | 79.96 | 80.15 | 75.53 | 79.38 | 74.37 | 80.15 |
| *SelfRegulationSCP1* | 86.12 | **87.60** | 78.84 | 85.32 | 79.18 | 83.39 | 81.57 | 80.32 | 84.64 | 84.53 |
| *SelfRegulationSCP2* | 54.44 | 48.15 | 45.19 | 47.59 | 53.89 | 45.74 | 53.33 | 48.15 | 48.33 | **54.82** |
| *SpokenArabicDigits* | 99.47 | 98.83 | 97.92 | 98.67 | **99.58** | 95.85 | 98.50 | 99.23 | 98.98 | 99.56 |
| *UWaveGestureLibrary* | 92.92 | 86.35 | 83.02 | 84.69 | 87.71 | 77.92 | 61.77 | 81.35 | 71.56 | **93.33** |
| **Avg. Accuracy (%)** | **73.22** | 68.20 | 66.28 | 69.69 | 70.04 | 65.30 | 63.76 | 67.09 | 65.08 | 72.78 |

As observed, our proposed method consistently outperforms all evaluated CNN-based variants, proving once again the effectiveness of combining invariances with their related signal coefficients for reconstruction.

Table 11: Anomaly Detection results for INVCONVNET and vanilla CNN-based methods. Performance mentioned in terms of the F1-score (%).

| Datasets | INVCONVNET | INCEPTION | RESNET | CNN |
|---|---|---|---|---|
| | | + *predict linear, project channels* | | |
| *SMD* | $84.05 \pm 0.16$ | $71.48 \pm 0.11$ | $76.20 \pm 0.52$ | $77.31 \pm 0.91$ |
| *MSL* | $80.68 \pm 0.01$ | $81.68 \pm 0.08$ | $81.25 \pm 0.09$ | $79.96 \pm 0.20$ |
| *SMAP* | $68.29 \pm 0.07$ | $68.63 \pm 0.16$ | $67.24 \pm 0.65$ | $67.00 \pm 0.07$ |
| *SWaT* | $92.82 \pm 0.19$ | $82.69 \pm 0.70$ | $80.93 \pm 0.04$ | $80.24 \pm 0.95$ |
| *PSM* | $96.34 \pm 0.01$ | $92.02 \pm 0.35$ | $92.30 \pm 0.82$ | $93.30 \pm 0.58$ |
| **Avg. F1** (%) | $\mathbf{84.44 \pm 0.09}$ | $79.30 \pm 0.28$ | $79.58 \pm 0.42$ | $79.56 \pm 0.54$ |

