# OpenReview forum: "Invariant Convolutional Layers for Time Series"
_ICLR.cc/2025/Conference — Submitted to ICLR 2025_

### Official Review · Reviewer_Ao1z · 2024-10-23

**Soundness:** 3
**Presentation:** 3
**Contribution:** 3
**Rating:** 6
**Confidence:** 2

**Summary:**

This paper introduces a new type of convolutional neural network layer, invariant convolutional layers, which are designed for time series data. The layer can capture invariant features by taking into account common deformations in time series, such as shifts, scaling, and linear trends. This paper proposes an architecture called INVCONVNET, which combines standard and invariant convolution to improve the performance of time series classification and anomaly detection. The experimental results show that INVCONVNET outperforms existing baseline methods on multiple datasets with better generalization and computational efficiency.

**Strengths:**

This paper contains a large number of mathematical proofs, showing that convolution can capture time series invariance, and sufficient experiments in time series classification and anomaly detection demonstrate the importance of the proposed invariant convolution.

**Weaknesses:**

The INVCONVNET model presented in the paper, while excellent at time series classification and anomaly detection tasks, may have some limitations.Among the existing methods for capturing time series invariance, linear models and frequency-domain models （Dlinear and FITS) perform well and are not compared with them in this paper.

**Questions:**

1.The time series itself contains variable features and time invariant features, so why design only for time invariant features?
2.As the conversion of time series to frequency domain proves to be a suitable method for capturing time-invariant features, why does this paper not consider converting time series to frequency domain for learning by convolution first?
3. I find that there are many research about Time-invariant convolution,please further clarify the contribution of this article
[1]. Chaman, A., & Dokmanic, I. (2021). Truly shift-invariant convolutional neural networks. In Proceedings of the IEEE/CVF Conference on Computer Vision and Pattern Recognition (pp. 3773-3783).
[2].Chen, X., Cheng, Z., Cai, H., Saunier, N., & Sun, L. (2024). Laplacian convolutional representation for traffic time series imputation. IEEE Transactions on Knowledge and Data Engineering.

---

> ### Author Response · Authors · 2024-11-19
>
> We would like to sincerely thank the reviewer for acknowledging the good level of our contribution in terms of methodological soundness, mathematical presentation, and significant performance improvements. We are pleased to address the concerns raised by the reviewer by carefully replying to the provided comments and questions below.
>
> - **[Reply to Weakness - Comparison with Linear and Frequency-Domain Models:]** MLP-based models and models that extract frequency-domain features have indeed recently proven quite successful in time series modeling, with several relevant architectures that we thoroughly present in the provided related work section (please see the first paragraph of Section 2). In terms of comparisons for both classification and anomaly detection, we leverage 2 state-of-the-art frequency-based deep learning architectures, namely TimesNet and TSLANet, that capitalize on Fast Fourier Transform to extract frequency-domain representations to capture multiple periodicities and achieve robustness to noise, respectively. Both frequency-based TimesNet and TSLANet methods are experimentally motivated as general-purpose convolutional time series modeling methods which explains our choice to include them as baselines for both classification and reconstruction-based anomaly detection. On the contrary, the FITS method proposed by the reviewer, while built upon frequency-domain features, is originally designed and motivated for time series regression tasks, namely forecasting, and anomaly detection. In this direction, we have already considered in our study several successful models in time series regression tasks applied to time series classification (we refer to TimesNet, TSLANet, DLinear, PatchTST, Crossformer baselines) and even more in number for time series anomaly detection (we refer to TimesNet, TSLANet, DLinear, LightTS, PatchTST, ETSformer, FEDformer, Autoformer, Pyraformer, Informer, Reformer). With respect to the MLP-based baseline DLinear that is suggested by the reviewer, we would like to clarify here that it is already included as a baseline in both classification and anomaly detection experiments and significantly outperformed the proposed method, and several baselines, especially for the CNN-dominated task of time series classification. Based on the above justifications, we believe that the considered baselines include well-established and widely cited approaches, reflecting a broad spectrum of effective building blocks (such as convolutional, attention, and linear but also frequency-based ones) from the relevant literature.
>
> - **[Reply to Q1 - Layer Design for Variant and Invariant Features:]** We think there is a potential misunderstanding in this comment. Our proposed convolutional layer is not built solely upon invariant filters. In the methods section 3, paragraph **pool of convolutions**, we indeed provide mathematical formulations for the design of convolution layers, including filters of increasing order of invariance complexity. In our experiments, we considered a pool of convolution layers concatenating three types of filters, which are (i) normal ones (similar to vanilla convolutional filters), (ii) filters invariant to offset shift and amplitude scaling, and (iii) filters invariant to offset shift, linear trend and amplitude scaling.
> Based on this, our proposed convolutional layer is built upon both variant and invariant filters in order to capture both variant and invariant features of the time series.

---

> ### Author Response · Authors · 2024-11-19
>
> - **[Reply to Q2 - Learning Invariant Features in the Frequency Domain:]**
> As the reviewer pointed out, learning invariance in the frequency domain is also a suitable approach for time series. Models operating in the frequency domain typically decompose a signal into its Fourier basis, apply a filter via point-wise multiplication, and then transform the signal back to the temporal domain. However, applying such filters directly to the decomposed signal inherently leads to the learning of global invariance rather than the local invariance we address with our framework. To mitigate this issue, methods like TSLANet [1] segment the signal and apply filters in the frequency domain to each segment. While this limits the filters to specific frequencies due to the discrete Fourier transform (DFT), it raises the challenge of capturing local invariant features in the temporal domain while maintaining sufficient model discriminativeness. This trade-off can be understood through the uncertainty principle, which governs the resolution trade-off between time and frequency domains. Our approach addresses this challenge by operating entirely in the temporal domain. This allows us to focus on extracting local features invariant to deformations, which require a broad spectrum of frequencies to be encoded. As demonstrated in our experiments (Section 4), this distinction leads to both performance and computational advantages. Our lightweight network (200K parameters for an example dataset in Figure 3) achieves top performance (0.72 accuracy on the UEA dataset, as shown in Table 1 and Figure 3) by encoding invariance to real-world deformations such as amplitude, offset, and trend. In contrast, TSLANet [1], a significantly larger network (3M parameters), achieves lower performance (around $69\%$ average accuracy on UEA).
>
> - **[Reply to Q3 - Relevant Works on Invariant CNNs and Shift Invariance:]** We would like to refer the reviewer to our General Comments section and specifically **[Comment 1]**, where we provide justifications about the positioning of the work with respect to works on invariant CNNs. In the original draft, we had originally added this discussion in the introductory Section 1, page 2, in the second paragraph. We are willing to change this paragraph if the updated discussion addresses the expressed concerns. Regarding the referenced papers [2, 3], the first one [2] deals with shift invariance in image processing. While convolutions are inherently equivariant to shifts, achieving shift invariance typically involves ending convolution layers with an average pooling layer, ignoring boundary effects. However, most current architectures insert downsampling layers between convolution layers, which introduces sensitivity to shifts. In [2], the authors address this issue by proposing a downsampling method that maintains shift invariance. Similarly, our proposed invariant convolutions are also equivariant to time shifts, and networks using these convolutions can achieve time-shift invariance depending on their structure. For example, in our experiments, all networks are time-shift invariant because we conclude the convolution layers with average pooling without incorporating downsampling layers. The second paper [3] addresses data imputation in traffic time series. Importantly, time series have a meaningful sequential structure where the order of observations is critical. However, traditional methods for data imputation based on low-rank matrices are invariant to permutations of rows and columns, which leads to poor performance on time series data. To address this limitation, the authors proposed a novel imputation method that leverages convolution to ensure local dependence on time ordering, thereby counteracting permutation invariance. This approach differs significantly from ours, as their work focuses on overcoming irrelevant invariance in existing imputation methods by exploiting time series structures. In contrast, our work focuses on encoding meaningful invariance for time series within convolutions.
>
>
> [1] Eldele, E., Ragab, M., Chen, Z., Wu, M., & Li, X. (2024). Tslanet: Rethinking transformers for time series representation learning. arXiv preprint arXiv:2404.08472.
>
> [2] Jacques, B. G., Tiganj, Z., Sarkar, A., Howard, M., & Sederberg, P. (2022, June). A deep convolutional neural network that is invariant to time rescaling. In International conference on machine learning (pp. 9729-9738). PMLR.
>
> [3] Chen, X., Cheng, Z., Cai, H., Saunier, N., & Sun, L. (2024). Laplacian convolutional representation for traffic time series imputation. IEEE Transactions on Knowledge and Data Engineering.

---

> > ### Comment · Reviewer_Ao1z · 2024-11-22
> >
> > I think the author has better answered my questions, so I will change the score to 6

---

### Official Review · Reviewer_vw6U · 2024-10-28

**Soundness:** 3
**Presentation:** 3
**Contribution:** 2
**Rating:** 6
**Confidence:** 4

**Summary:**

This paper introduces a novel approach to handling time series data by developing convolutional layers that are invariant to specific types of deformations commonly found in time series, such as offset shifts, scaling, and linear trends. The author mathematically formulate invariances for time series using group theory, designed invariant convolutions that can handle these deformations. The effectiveness of the proposed method is empirically validated through classification and anomaly detection tasks.

**Strengths:**

1. The overall paper is clearly written and easy to understand, the paper structures are well organized.

2. The mathematical formulation standardize the invariant definition for certain time series properties

3. Efficient implementation of invariant convolutions using FFT

**Weaknesses:**

1.	My major concern is the gap from theoretical definition of time series invariants to the actual empirical improvement on time series tasks. For CNN base time series model, the major factor to improve classification task performance is tuning multi-scale convolution kernel sizes (check paper: https://arxiv.org/abs/2002.10061). This reviewed paper seems also using this “Pool of convolutions” (line 302). Without more detailed ablation studies for each of the components, it is hard to justify where the actual contribution come from.

2.	Following the above discussion, for time series classification tasks, the SOTA models are still cnn based not transformer based. Therefore, it is more meaningful to use cnn based time series models such as TimeNet or Omni-scale CNN as baselines instead of transformer based models.

3.	The invariant convolution layer implementation is not very clear. Upon checking the appendix, there still lacks a detailed description of neural network architectures such as layer number/ kernel size for each specific layer.

**Questions:**

1.	How does the choice of kernel sizes affect the model's ability to capture different types of invariances? Is there a systematic way to select optimal kernel sizes based on understanding of “invariants”?

2.	Can you show some quantitative results that the computation efficiency gain of FFT-based convolution layer over classical kernel with longer time series?

3.	How is the proposed new invariant feature linked to/different from some standard normalization treatments in ML? i.e. the example in line 256 looks like very similar to kernel normalization.

---

> ### Author Response · Authors · 2024-11-19
>
> We really appreciate the time devoted by the reviewer to provide very interesting comments on our contribution and would like to thank him for highlighting the clear presentation, mathematical soundness, and efficient implementation of our proposed Invariant Convolutional Layers. In response to the issues raised by the reviewer, we address each point in detail.
>
> - **[Reply to W1 - Pool of Convolutions:]**
> In the methods section 3, we provide the mathematical formulations for the design of a novel convolutional layer that can be invariant to a pool of predefined deformations. In our experiments, the pool of convolutions corresponds to the concatenation of three types of filters in our proposed convolutional layer, which are (i) normal ones (similar to vanilla convolutional filter), (ii) filters invariant to offset shift and amplitude scaling, and (iii) filters invariant to linear trend and amplitude scaling. The formalization for the pool of convolutions is provided in the relevant paragraph on page 6, and its components are visualized in Figure 2 (Right). The term "pool of convolutions" in our case, deviates from the one in the paper provided by the reviewer [3], in the fact that we refer to the operation of combining different types of filters with respect to different invariances. Combining the output of vanilla (non-invariant) convolutional filters (e.g., by concatenation), which is performed in [3], is rather a very common practice for extracting features corresponding to different receptive fields based on the characteristics of the filter (e.g., kernel size, dilation). For instance, this is also common in the classical Inception architecture, which we use as a baseline, among several others.
>
> - **[Reply to W1 - Components Contribution in Invariant Convolutional Layer:]** We already provide ablation studies for the component's contribution to our proposed convolutional layers, with respect to the potential influence of the types of invariances in multivariate classification for all considered datasets (26 datasets of UEA, UCIHAR, Sleep-EDF, Epilepsy) in Table 2 and additional results and for the synthetic experiment on the 4 larger univariate UCR datasets in Table 3 (see also [Comment 2] of the General Comments section for [Additional Results] provided for a stronger synthetic deformation). For the ablation studies on the components performance, we consider filters that correspond to (i) mixed types of filters (model named InvConvNet), (ii) solely normal filters (no invariance, model named InvConvNet-N), (iii) filters invariant to offset shift and amplitude scaling (model named InvConvNet-O), and (iv) filters invariant to linear trend and amplitude scaling (model named InvConvNet-T). Our proposed layer is built upon mixing the three types of filters in terms of invariance (denoted as the pool of convolutions). In the ablation study, we consider the same optimal total number of hidden dimensions, which is derived by tuning a hyperparameter denoted as $d$. For the mixed types of invariances, each type of convolution receives almost one-third of the $d$ dimension, with adjustment to sum up to $d$ (more details are explained in the hyperparameter selection paragraph of Section Appendix A.4).
>
> - **[Reply to W1 - Comparisons with Multi-Scale CNNs:]**
> As mentioned by the reviewer, capturing information at different scales with CNN is a common approach for time series, which has proved to be relevant in numerous cases. Typically, this is achieved either by increasing the network's depth or by combining kernels of different sizes, as in Omni-scale [3]. Among the considered baselines, several are built upon stacked convolutions or different kernel sizes to capture multiple scales, such as TSLANet, Inception, ResNet and Rocket.
> In our manuscript, we formally introduce the invariant convolution for arbitrary kernel sizes. In the experiments, we incorporate kernels of three different sizes to remain competitive with other methods, which also employ multi-scale strategies. It is worth mentioning that we did not focus on fine-tuning the kernel sizes, as they were determined heuristically (see Appendix A.4). The critical question of whether combining invariance with CNNs provides improvements independent of multi-scale strategies is addressed in our ablation study (Section 4, page 8). On the UEA datasets, we observed a $3\%$ increase in accuracy when incorporating invariant convolutions of various types into CNNs. This result demonstrates that adding invariance to convolutions enhances the performance of CNNs for classification tasks.

---

> ### Author Response · Authors · 2024-11-19
>
> - **[Reply to W2 - Choice of Baselines/Inclusion of CNN methods:]** There is potentially a misunderstanding for the reviewer with respect to the type of neural network architectures used as baselines for time series classification. More specifically, as shown in Table 1, we use 9 baselines in total, among which 6 are convolutional-based (namely TimesNet, TSLANet, Inception, ResNet, CNN, and Rocket), 2 are built upon attention-forming transformer-like architectures (i.e., PatchTST and Crossformer), and 1 is a purely linear architecture (DLinear). Additional transformer-based architectures were used as baselines for the task of reconstruction-based anomaly detection, following the recent trends in time series regression tasks, but definitely this is not the case for the experimental design we followed in classification. Additionally, among the two self-supervised contrastive learning methods evaluated in the Transfer Learning experiment, TS-TCC is built upon transformer-like modules, while TS2Vec is built upon stacked dilated convolutions interleaved with max-pooling operations. Based on the above, the vast majority of the considered baselines for the classification tasks are indeed convolutional, yet a wide variety of building modules has been tested for the different setups, always following the trends of relevant papers in this field.
>
> - **[Reply to W3 - Details of Hyperparameters in InvConvNet Design:]** We would like to ask the reviewer about the exact part that could potentially benefit from more details on the hyperparameter selection for the employed Invariant Convolutional Layers inside the example InvConvNet architectures. Explicit details about the ranges in which hyperparameters, such as the kernel size and the number of hidden dimensions, were chosen are explicitly stated in Appendix A.4, 3rd paragraph, page 19, lines 995-1006. Specific combinations for each dataset are provided in scripts inside the code of the supplementary material for InvConvNet and the baselines used. The example InvConvNet architectures we leveraged to showcase the representational capabilities of the proposed Invariant Convolutional Layers are thoroughly explained in Appendix A.2.1 and Figures 4 and 5. Among the three different tested variants, two are single layer (in terms of depth - depicted in Figure 4(Left)), the so-called standard InvConvNet (1 kernel size), and the inception-like InvConvNet (3 distinct kernel sizes operated in parallel). Both single-layer variants excel in the datasets used for classification, as mentioned in Appendix A.4. Finally, one InvConvNet variant, the so-called multi-scale (depicted in Figure 4(Right)), is a 2-layer to capture finer granularity, which is crucial for the reconstruction-based anomaly detection task. The first layer is the proposed convolutional layer built upon a pool of convolutions (of kernels that are (i) variant, (ii) invariant to amplitude scaling and offset shift, and (iii) invariant amplitude scaling and linear trend invariant) and the second layer is a standard 1D-CNN (details are given in the appendix).

---

> ### Author Response · Authors · 2024-11-19
>
> - **[Reply to Q1 - Effect of Kernel Size in Capturing Invariances:]** The kernel size of the proposed convolutional layer is indeed an interesting aspect for further discussion. More specifically, we can intuitively assume and experimentally confirm that a kernel, in order to have sufficient information to extract the designed amplitude-related invariances, needs points that are at least equal to a minimum value between 50 and and half of the length of the time series (which is also mentioned in Appendix A.4 in hyperparameter selection). This choice of the kernel length, while rather heuristic, seems to apply with success to the wide range of the tested time series datasets. Additionally, training larger kernels in our layers is achieved by the Fast Fourier Transform, as mentioned in Section 3.2, page 6. On the contrary, leveraging very small kernels, e.g., of length 4 or 8, that are commonly employed in standard convolutions could not be sufficient for learning robust representations incorporating invariances with respect to amplitude scaling. Finally, like standard convolutions, very large kernels can be inefficient in training.
>
> - **[Reply to Q2 - Computational Efficiency of FFT-based over Standard Convolution:]**
> As thoroughly explained in the methods Section 3.2, page 6, paragraph on the fast computation of our proposed convolutional layers, we leverage the Fast Fourier Transform (FFT) for performing the proposed convolutions [1] to enable the use of larger kernel sizes (unlike standard CNN-based models that typically use small kernel sizes of 2 or 4 elements in the convolution operation). Based on this fast implementation, the computation is not affected in terms of time cost by the kernel size. While several implementations can be publicly found inside the repository [2] from which we derived the code for the FFT computation of the convolution, there is a time complexity comparison for the execution time with increasing kernel sizes for standard (in blue) and FFT-based convolution (in red). This plot confirms the time efficiency of FFT-based convolution for increasing kernel sizes. For $N$ output channels, the complexity of the operations using standard convolutions (non-FFT-based) is $\mathcal{O}(BNCWL)$, where $B$ is the batch size, $L$ is the length of the series, $C$ the channels and $W$ the width of the kernel size, whereas for the proposed convolutions (FFT-based) as described above and in the relevant section in the manuscript, the complexity of the computations is $\mathcal{O}(BNCL\log (L))$. Therefore, for $\log(L) < W$, the proposed convolutional layer is more efficient than standard 1D-CNN.
>
> - **[Reply to Q3 - Invariant Convolution for Time Series Compared to Normalization Techniques:]**
> The z-normalization discussed in the example on page 5 is a well-established technique in time series data mining [4]. It produces an embedding that is invariant to amplitude scaling and offset shifts, a property that is naturally derived within our mathematical framework. In the deep learning community, z-normalization is commonly associated with the batch normalization layer, which is designed to accelerate and stabilize the learning process.
>
> However, there is a fundamental distinction between batch normalization and our approach. Batch normalization operates at a global scale, computing invariance for the entire time series. In contrast, the proposed invariant convolution computes invariance at a local scale, corresponding to the size of the convolutional kernel. This means that our method captures features that are locally invariant to well-designed time series deformations.
>
> [1] Mathieu, M., Henaff, M., & LeCun, Y. (2013). Fast training of convolutional networks through ffts. arXiv preprint arXiv:1312.5851.
>
> [2] Odom, F. III. (2023). fkodom/fft-conv-pytorch [Software]. Plainsight. Retrieved from https://github.com/fkodom/fft-conv-pytorch
>
> [3] Tang, W., Long, G., Liu, L., Zhou, T., Blumenstein, M., & Jiang, J. (2020). Omni-scale cnns: a simple and effective kernel size configuration for time series classification. arXiv preprint arXiv:2002.10061.
>
> [4] Paparrizos, J., Liu, C., Elmore, A. J., \& Franklin, M. J. (2020, June). Debunking four long-standing misconceptions of time-series distance measures. In Proceedings of the 2020 ACM SIGMOD international conference on management of data (pp. 1887-1905).

---

> > ### Comment · Reviewer_vw6U · 2024-11-23
> > **reply to author rebuttal**
> >
> > I thank the authors for answering my questions, and my concerns are mostly addressed. I will raise to 6.

---

### Official Review · Reviewer_akRu · 2024-11-01

**Soundness:** 2
**Presentation:** 1
**Contribution:** 2
**Rating:** 3
**Confidence:** 4

**Summary:**

The paper tackles the challenge of developing invariant convolutional layers tailored for time series. It begins by introducing a mathematical framework for invariant convolution in time series, followed by the design of specific architectural blocks that ensure invariance to offset shifts, scaling, and linear trends. These invariant blocks are then integrated with convolutional layers. The proposed approach is evaluated through experiments on both univariate and multivariate time series, covering tasks such as classification and anomaly detection.

**Strengths:**

- S1. The paper addresses the compelling problem of designing invariant deep learning models for time series, which is valuable for tasks where invariance to certain properties, such as offset, is essential. This is particularly relevant in applications like classification, where only specific patterns are informative.

- S2. Additionally, the paper explores a relatively uncharted area of geometrical deep learning for time series, providing valuable insights into this underdeveloped research area.

- S3. The appendices are thorough and greatly enhance the reader’s understanding of the model architecture and the processes used in the supervised tasks, especially Appendix A.2 ("InvConvNet: Architectural Details"), which contains essential architectural details.

- S4. The experimental evaluation is comprehensive, covering two key supervised tasks (classification and anomaly detection) and providing a broad perspective on the model’s performance.

**Weaknesses:**

- W1. The positioning of the work is somewhat unclear, as it focuses exclusively on specific invariance properties for time series without connecting to the broader research in convolutional neural network invariances. Invariance is a well-studied area in deep learning (e.g., CNN shift-equivariance, see [1]), and discussing this literature could enrich the context and impact of the paper.
- W2. The paper is challenging to follow:
    - The extensive two-page mathematical formalism on invariant embeddings for time series may not aid reader comprehension as intended.
    - The architecture description lacks clarity; key elements are dispersed and essential components are placed in the appendices (such as in Appendix A.2), making it difficult to grasp the model’s structure. Reorganizing and detailing the architectural explanations would enhance readability.
    - Consider moving the content of Appendix A.2 to the main body and placing the formalism of invariant embeddings in the appendix to streamline the presentation of contributions.

- W3. The experimental results are not stellar. In the classification tasks, the InvConvNet model does not significantly outperform ResNet and Rocket on the UEA archive datasets (Table 1 and Table 9). Furthermore, on four selected UCR datasets, InvConvNet performs worse than its variant without invariant blocks (Table 3), despite showing some robustness to deformation.
- W4. The practical value of the proposed InvConvNet architecture is unclear. It may be beneficial to focus on specific datasets where invariance to offset and trend is particularly desirable, which could demonstrate the model's relevance in applied scenarios.

**Questions:**

- Q1. The description of the Transfer Learning Experiment is unclear. Could you provide a more detailed explanation of this experimental setup ?

- Q2. Could you please discuss other invariance like shift invariance?

- Q3. Could you address the concerns raised in the weaknesses section ?

---

> ### Author Response · Authors · 2024-11-19
>
> We truly thank the reviewer for the time devoted to assessing our work and providing interesting comments. Additionally, we are pleased to hear that the reviewer confirms the novelty of the proposed architectural design and the significance of the paper's overall research direction, as well as the presentation of key components of the model architecture and the presented experimental setups. Following the provided comments, we next aim to address any weaknesses and questions raised by the reviewer.
>
> - **[Reply to W1 - Relevant Works on Invariant CNNs:]** We would like to refer the reviewer to our General Comments section and specifically **[Comment 1]**, where we provide justifications about the positioning of the work with respect to works on invariance in deep learning. In the original draft, we had originally added this discussion in the introductory Section 1, page 2, in the second paragraph. We are willing to change this paragraph if the updated discussion addresses the expressed concerns.
>
> - **[Reply to W2 - Math Details vs Architectural Details in Methods Section:]** We would like to thank the reviewer for their thorough suggestions and the time invested in providing comments relevant to the structure and presentation of the paper. However, we would like to highlight that the central contribution of our work (also reflected in the title) is the formalism and mathematical design of fast convolutional layers that are invariant to specific group actions targeted to the relatively underexplored field of time series invariance in deep learning. We, therefore, believe that the mathematical details, as also confirmed by many of the other reviewers, remain a significant part of the presented work in terms of potential impact and positioning against the vast abundance of explicit architectural designs for time series tasks. Contrary to this, our layers are general and could be adapted to a time series model as feature extractors to leverage invariances. For this reason, we chose to present examples of architectures that are shallow (see Figure 4) and simple yet achieve robust performances against complex baselines. Due to the page limit, we kept the task-specific architectural details in the appendix yet tried to present key aspects in the last paragraph of the methods (section 3) and the experimental setups described in the main paper. We are pleased to hear the reviewer's opinion on the above arguments about the chosen structure of the methodological section and potentially address specific parts that could potentially enhance the coherence of the architectural design in the paper.
>
> - **[Reply to W3 - Significance of Overall Results]** We kindly disagree with the reviewer on the point made for the significance of the overall results achieved by the proposed invariant convolutions. This is clearly shown in the classification experiments on UEA and additional datasets (UCIHAR, Epilepsy, Sleep-EDF) with accuracy improvements up to $0.7\%$ compared to the best competitor out of several quite complex and computationally demanding baselines (an example is shown in Figure 3).
> When adding realistic deformations such as offset and trend to simple UCR datasets, the drop in classification performances of the proposed InvConvNet with mixed convolution types (including invariant ones) remains relatively low compared to standard convolutions (please see Table 3 and [Comment 2] of the General Comments section for additional comments/results). It demonstrates that standard CNNs have difficulties to correctly learn invariant features which corroborates with results observed in [1] for image data. In the transfer learning experiment (please see [Comment 3] of the General Comments section for additional comments/results), when evaluating an unseen target domains, the proposed InvConvnet offers average improvements of around $4\%$ compared to contrastive learning methods that leverage general-purpose self-supervised pretraining. Finally, on the more demanding reconstruction-based task of anomaly detection where the need for scale invariances can be questioned, the proposed InvConvNet architecture remains relatively competitive.
>
> - **[Reply to W3 - Inconsistency in the UCR Ablation Study of Table 3:]** We refer the reviewer to our answer in [Comment 2] of the General Comments section for comments/additional results. The slightly higher accuracy achieved by InvConvnet-N (built on only normal convolutions) can be attributed to the inherent z-normalization of the employed UCR datasets and the simplicity of the classification. In contrast, when synthetic deformations with respect to offset and linear trend are added, our proposed and mixed InvConvNet achieves better performances than the variants built solely upon one type of convolution.

---

> ### Author Response · Authors · 2024-11-19
>
> - **[Reply to W4 - Practical Value of Invariant Convolutions:]** The practical value of the proposed Invariant Convolutions is the formalization of convolutional layers that are invariant to group actions specifically designed for time series data, which are incorporated explicitly in the layer design but offering flexibility into learning these properties. Standard neural network architectures, when applied to noisy and non-stationary time series data, often exhibit generalization issues and can be prone to overfitting. This is clearly showcased in the example of the ECG data in Figure 1, where a standard convolution fails to capture relevant features of the signal. Incorporating knowledge on learning invariant features for time series is often done by applying strong augmentations external to the model architecture (for instance, self-supervised contrastive learning forces positive and negative views by employing targeted loss function in training). Yet, the way augmentations are produced increases the computational complexity and makes strong assumptions about the data distribution (mentioned in several contrastive learning time series works). For instance, our method, while not depending on unsupervised pertaining yet, excels when evaluated on different target data (please see Table 4). To the best of our knowledge, this is the first work in the field of time series that mathematically formalizes layers that are invariant to specific group actions with respect to amplitude scaling while providing theoretical guarantees. The proposed layer is incorporated in lightweight (see Figure 3) and shallow modules (see Figure 4), forming the example InvConvNet architecture, which is yet competitive in performance against very complex and deep architectures for time series modeling. The proposed Invariant Convolutions can act as a solid mathematical tool and strong feature extraction module in future works that aim to study robust models for time series analysis.
>
> - **[Reply to Q1 - Details on the Transfer Learning Experiment:]** We refer the reviewer for [Comment 3] of the General Comments section for additional details/results on the Transfer Learning Experiment. The self-supervised methods are pre-trained and fine-tuned on each source domain and evaluated on the unseen target domains. The supervised method InvConvNet is trained on the source domain (in a supervised way) and evaluated on the unseen target domains. Interestingly, even with the absence of general-purpose representations agnostic to the task, our proposed invariant convolutions are able to learn features transferable to the unseen target domains, which can be attributed to the successful extraction of invariant features through our specifically designed layers.
>
> - **[Reply to Q2 - Discussion on Shift Invariance in CNNs:]** As noted in **[Comment 1]** of the General Comments Section, Reviewer WiU6 invited us to compare our contribution with other works [5, 6], which addresses time scale and shift-invariance in deep learning, respectively. Below, we outline the specific differences between these works and our approach. The first paper [5] addresses invariance to the group of time scalings defined as  $( t \mapsto at \ | \ a \in ]0, +\infty[ )$. Our work diverges in several ways. First, we address spatiotemporal invariance rather than solely time invariance. Secondly, the greedy approach in [6] approximates time scaling invariance by applying convolutions at multiple logarithmic scales and selecting the optimal scale through max pooling. This results in approximate invariance. In contrast, our mathematical framework provides an exact formulation for invariant convolutions, which can be computed efficiently using the FFT. The second paper [6] deals with shift invariance in image processing. While, as you mentioned, convolutions are inherently equivariant to shifts, achieving shift invariance typically involves ending convolution layers with an average pooling layer, ignoring boundary effects. However, most current architectures insert downsampling layers between convolution layers, which introduces sensitivity to shifts. In [5], the authors address this issue by proposing a downsampling method that maintains shift invariance. Similarly, our proposed invariant convolutions are also equivariant to time shifts, and networks using these convolutions can achieve time-shift invariance depending on their structure. For example, in our experiments, all networks are time-shift invariant because we conclude the convolution layers with average pooling without incorporating downsampling layers.

---

> ### Author Response · Authors · 2024-11-19
>
> [1] Kvinge, H., Emerson, T., Jorgenson, G., Vasquez, S., Doster, T., & Lew, J. (2022). In what ways are deep neural networks invariant and how should we measure this?. Advances in Neural Information Processing Systems, 35, 32816-32829.
>
> [2] Eldele, E., Ragab, M., Chen, Z., Wu, M., Kwoh, C. K., Li, X., \& Guan, C. (2021). Time-series representation learning via temporal and contextual contrasting. arXiv preprint arXiv:2106.14112.
>
> [3] Lessmeier, C., Kimotho, J. K., Zimmer, D., \& Sextro, W. (2016, July). Condition monitoring of bearing damage in electromechanical drive systems by using motor current signals of electric motors: A benchmark data set for data-driven classification. In PHM Society European Conference (Vol. 3, No. 1).
>
> [4] Ragab, M., Chen, Z., Wu, M., Li, H., Kwoh, C. K., Yan, R., \& Li, X. (2020). Adversarial multiple-target domain adaptation for fault classification. IEEE Transactions on Instrumentation and Measurement, 70, 1-11.
>
> [5] Jacques, B. G., Tiganj, Z., Sarkar, A., Howard, M., \& Sederberg, P. (2022, June). A deep convolutional neural network that is invariant to time rescaling. In International conference on machine learning (pp. 9729-9738). PMLR.
>
> [6] Chaman, A., \& Dokmanic, I. (2021). Truly shift-invariant convolutional neural networks. In Proceedings of the IEEE/CVF Conference on Computer Vision and Pattern Recognition (pp. 3773-3783).

---

> ### Comment · Reviewer_akRu · 2024-11-21
> **Response to the authors**
>
> I would like to thank the authors for their detailed responses and clarifications. While I will avoid reopening a lengthy discussion, I have a few additional remarks to share:
>
> - **UCR datasets.** I remain unconvinced by the explanation regarding the UCR dataset results, particularly the performance gap between InvConvNet and InvConvNet-N (-normal) in the setting without deformations. Z-normalization is a widely recognized standard technique when applying neural networks to time series classification. Therefore, I would expect the proposed InvConvNet to match the performance of InvConvNet-N (-normal) on the UCR dataset if it is to be considered competitive.
> - **Transfer learning experiments.** If I understand correctly, both contrastive representation methods, TS2Vec and TS-TCC, are trained in an unsupervised manner on the source datasets, and their classification head (an SVM in the case of the original TS2Vec paper) is also trained on the source datasets. The comparison then evaluates the zero-shot performance without retraining the classification head on the target dataset. This approach seems unusual, as it disregards the common practice of training the classification head on the target dataset for transfer learning tasks. This is especially important when the classification head functions as a geometric separator. Typically, in transfer learning experiments for classification (e.g., the T-Loss paper), the unsupervised architecture is trained on a source dataset and used as a universal feature extractor for target datasets, where the classification head is subsequently trained on the target representations.
> - **Mathematical framework.** As noted in my initial review, the general mathematical framework presented in Section 3.1 provides a helpful unification of certain invariant properties that may be desirable in specific time series modeling tasks. However, I remain unconvinced about the general usefulness of hard invariances such as linear trend invariance or amplitude scaling invariance in most time series classification scenarios.
> - **Discussion of invariance properties in Deep Learning.** Thank you for addressing the topic of invariance in deep learning and referencing related works like the "Shift Invariance in CNNs" paper in Computer Vision. In my opinion, this broader discussion provides a valuable context for positioning the paper.
>
> I appreciate the effort in your rebuttal, but for the reasons stated above, I will keep my score the same.

---

> > ### Author Response · Authors · 2024-11-23
> >
> > We really appreciate the reviewer's comments and effort in assessing our work. While understanding that the reviewer does not intend to open a big discussion, we would like to justify some points in short in order to support the conception and presentation of our work.
> >
> > - **UCR datasets.** We respectfully disagree with the reviewer on the point: "expect InvConvNet to match the performance of InvConvNet-N (-normal) on the UCR dataset if it is to be considered competitive," and we next explain the reasons. The goal of our study on the $4$ UCR datasets was to provide a comprehensive ablation of the main components of the model (variant and invariant ones) on synthetic deformations. The performance on the plain (i.e., z-normalized input) is matched for 2 out of 4 plain UCR considered datasets (and is greater for the added deformations), so it is not the case that our model does not match the configuration solely made upon normal convolutions (InvConvNet-N (-normal)). This is also clearly shown in several experiments (Table 1, Table 2, updated Table 4 in the experiment in General Comment 3 of the rebuttal), where our method outperforms not only InvConvNet-N (-normal) but in several cases also the 6 considered CNN-based variants, which constitute very competitive frameworks in the field of time series. We were not concerned about the small performance gap of InvConvNet against InvConvNet-N (-normal) on the few plain UCR datasets, and this is why we chose to show the results with this specific subset of UCR datasets. Additionally, for ease of the setup in the different experiments, we gave an equal number of dimensions for our three distinct convolutional parts with respect to invariances while fine-tuning their contribution with different numbers of hidden dimensions per component could potentially show greater performances. Finally, z-normalization is a standard and effective pre-processing technique, particularly common for time series data mining, yet remains limited to removing the offset globally, which can be constraining/insufficient in some settings. On the contrary, our model learns to be locally invariant to offset shifts and linear trends, which proves quite effective for several datasets in our extensive experiments section.
> >
> > - **Transfer learning experiments.** We agree with the reviewer's comment about the common concept of fine-tuning the classification head on the target dataset in transfer learning experiments. However, as we already explained, we followed the guidelines given in the work of TS-TCC for the transfer learning experiment (without adaptation) on the Fault-Diagnosis dataset. We wanted to provide a fair comparison for our proposed example framework (built upon invariant convolutions), which is supervised, and the self-supervised methods. With the choice of direct transfer (no involvement of the target domain in training), we avoid giving an apriori advantage to the supervised method.  Additionally, we inspect that this proposed setup, particularly for Fault-Diagnosis datasets, is attributed to the fact that the source and target datasets share the same set of classes, and training/fine-tuning on the target domain experimentally achieves easily top accuracies for the target test sets. Thus, further training is commonly avoided in the target domains for the Fault-Diagnosis dataset.
> >
> > - **Mathematical framework.** Thank you for pointing out the usefulness of our mathematical framework. The practical contribution of the proposed layers and hard-coded invariances, in general, can be briefly summarized in (i) enhanced generalization and robustness to unseen data under the designed deformations, (ii) reduced model complexity (our examples architectures are lightweight, practices like data augmentation can be avoided), (iii) reduced data requirements in terms of volume or preprocessing. While we respect the reviewer's opinion, we tried to motivate and prove our proposed layer's advantages by evaluating them in example architectures across different datasets and settings.
> >
> > Thank you once again for your time and your engagement in the discussion.

---

> > > ### Comment · Reviewer_akRu · 2024-11-26
> > > **Comment**
> > >
> > > Thank you for your reply and the provided explanations.

---

### Official Review · Reviewer_WiU6 · 2024-11-03

**Soundness:** 2
**Presentation:** 3
**Contribution:** 2
**Rating:** 5
**Confidence:** 3

**Summary:**

The paper introduces INVCONVNET, a novel approach addressing time series analysis through mathematically-grounded invariant convolutions. The authors formulate a theoretical framework for handling common deformations in time series data, specifically targeting offset shifts, scaling, and linear trends. The key technical contribution lies in their construction of group action-based invariant operations, which they effectively combine with standard convolutions in a hybrid embedding layer. The work bridges theoretical foundations of invariant representations with practical architectural design, demonstrating applications in both classification and anomaly detection tasks. The authors provide mathematical formulations of their invariant convolutions and validate their approach experimentally against baseline methods.

**Strengths:**

1. Theoretical Foundation: The authors provide formal mathematical definitions of time series deformations in terms of group actions, creating a theoretically sound framework rather than just an empirical solution.
2. Structured and Explicit Design: Rather than relying on implicit learning of invariances through data augmentation or contrastive learning, they develop an explicit, structured approach to incorporating invariance properties directly into the architecture. This makes the solution more transparent and controllable.
3. Practical Problem with Clear Motivation: They demonstrate their approach through concrete examples like the ECG case, where standard convolutions fail with linear trends masking important patterns, showing clear real-world applicability.

**Weaknesses:**

- Fixed Kernel Bias: The initial motivation using a pre-defined kernel creates a biased comparison, without adequately addressing whether standard CNNs with learned kernels could naturally adapt to these deformations.
- Missing Comparative Analysis with Contrastive Learning: The authors don't compare their architectural approach against the simpler alternative of using these deformations (offset, trend) as augmentations in a contrastive learning framework, which could potentially achieve similar invariance properties with more flexibility.
- Incomplete Novelty Justification: The paper lacks proper positioning against existing relevant work on invariant architectures, particularly:
    - Jacques, Brandon G., et al. "A deep convolutional neural network that is invariant to time rescaling." *International conference on machine learning*. PMLR, 2022.
    - "Truly shift-invariant convolutional neural networks" A comparison with these works is crucial to establish the novelty and advantages of their approach.
- Contradictory Results to Core Motivation: The paper's fundamental premise is that invariance to deformations (offset, trend) is crucial for time series analysis. However, Table 3 directly contradicts this claim, showing InvConvNet-N (without invariance) performs best in 3 out of 4 clean datasets. This raises serious questions about the paper's core motivation and the necessity of architectural invariance.

**Questions:**

Check the Weakness Section

---

> ### Author Response · Authors · 2024-11-19
>
> We would like to thank the reviewer for their positive comments concerning the soundness of the mathematical details of the proposed Invariant Convolution, along with the definitions of time series deformations and group actions. We are also pleased to know that the reviewer recognizes the potential significant impact of our contribution to practical scenarios involving time series data, such as the presented ECG example in Figure 1. In the following responses, we aim to address each of the highlighted weaknesses individually and in detail to further support our contribution.
>
> - **[Reply to W1 - Fixed vs Learnable Invariant Kernel:]** In the introductory ECG example (Figure 1), we deliberately fixed the kernel to a single heartbeat and evaluated its correlation along an ECG signal affected by a trend. This example illustrates that a standard convolution fails to capture the high correlation between heartbeats, whereas a convolution that is locally invariant to linear trends successfully identifies these correlations. As noted in your review, this example effectively clarifies and motivates our work. For the sake of simplicity, we did not explore whether standard CNN layers can learn such invariance in the introduction. However, this question is explicitly addressed in the synthetic experiment (pages 8-9), where we analyze the decline in classification performance as baseline datasets with no distortions are progressively deformed with realistic synthetically generated deformations. The results demonstrate that standard CNNs struggle to learn invariances compared to invariant convolutions, aligning with findings in [1] for image data. A detailed description of the synthetic experiment is provided in **[Comment 2]** of the General Comments section. In the revised manuscript, we will explicitly acknowledge the limitations of the introductory example and indicate where this question is thoroughly addressed.
>
> - **[Reply to W2 - Comparisons to Augmentation-based Contrastive Learning Methods:]**
> We would like to grab the reviewer's attention to the Transfer Learning experiment presented in the relevant paragraph on page 9 (for performance scores, please see Table 4). In this experiment, we directly compare our proposed Invariant Convolutions (InvConvNet method) to two common self-supervised contrastive learning techniques that have shown great success in time series modeling, specifically TS-TCC [3] and TS2Vec [4]. This experiment involves training on a source dataset (left side of the arrow) and testing on a held-out target dataset (right side of the arrow). Self-supervised contrastive learning leverages self-supervised pre-training on data without labels, followed by supervised fine-tuning, which can give a significant advantage to the generalization properties of the backbone architectures. In our work, we carefully designed time series-specific invariant convolutions that can be incorporated into any supervised or unsupervised framework. Our proposed lightweight (see Figure 3) supervised InvConvNet acts as a simple proof of concept for the generalization capabilities of invariant convolutions. To be fair in comparison, since our example InvConvNet is supervised, we leveraged supervised baseline methods for classification and anomaly detection (Tables 1, 2, 3, 5). We applied the self-supervised contrastive methods to the Transfer Learning experiment, where they can show a natural advantage. More specifically, TS-TCC is indeed a transformation-based contrastive method that considers both weak and strong augmentations of the input. Weak augmentations involve adding random variations to the signal and scaling up its magnitude. In contrast, strong ones involve splitting the signal into a random number of segments and shuffling them. TS2Vec method criticizes the commonly used transformation-based consistency (scaling, jittering, permutations) in previous works for making strong assumptions on the data distribution. To overcome this, TS2Vec proposes contextual consistency, which treats representations at the same timestamp in two augmented contexts as positive and considers augmentations that do not change the magnitude of the data, such as timestamp masking and random cropping. Interestingly, those augmentations excel in classification performance in the original TS2Vec paper over common scale transformations, which even deteriorate performance when added to the method. We, therefore, chose two representative contrastive methods built upon different types of augmentations, including scale-level and timestamp-level ones. Our experimental results for the experiment in Table 4 showcase the excellent performance ($4\%$ avg. improvement) of invariant convolutions when transferred to unseen data over self-supervised techniques. We would also like to refer the reviewer for [Comment 3] of the General Comments section for additional details/results on the Transfer Learning Experiment.

---

> ### Author Response · Authors · 2024-11-19
>
> - **[Reply to W3 - Novelty Justifications and Relevant Works on Invariant Architectures:]** We would like to refer the reviewer to our General Comments section and specifically **[Comment 1]**, where we provide justifications about the positioning of the work with respect to works on invariance in deep learning. In the original draft, we had originally added this discussion in the introductory Section 1, page 2, in the second paragraph. We are willing to change this paragraph if the updated discussion addresses the expressed concerns. Regarding the referenced papers [5, 6], the first one [5] addresses invariance to the group of time scalings defined as  $( t \mapsto at \ | \ a \in ]0, +\infty[ )$. Our work diverges in several ways. First, we address spatiotemporal invariance rather than solely time invariance. Secondly, the greedy approach in [6] approximates time scaling invariance by applying convolutions at multiple logarithmic scales and selecting the optimal scale through max pooling. This results in approximate invariance. In contrast, our mathematical framework provides an exact formulation for invariant convolutions, which can be computed efficiently using the FFT. The second paper [6] deals with shift invariance in image processing. While convolutions are inherently equivariant to shifts, achieving shift invariance typically involves ending convolution layers with an average pooling layer, ignoring boundary effects. However, most current architectures insert downsampling layers between convolution layers, which introduces sensitivity to shifts. In [5], the authors address this issue by proposing a downsampling method that maintains shift invariance. Similarly, our proposed invariant convolutions are also equivariant to time shifts, and networks using these convolutions can achieve time-shift invariance depending on their structure. For example, in our experiments, all networks are time-shift invariant because we conclude the convolution layers with average pooling without incorporating downsampling layers.
>
> - **[Reply to W4 - Results on the UCR Ablation Study in Table 3:]** We thank the reviewer for pointing out this inconsistency, which could indeed benefit from additional justifications. We refer the reviewer to our answer in [Comment 2] of the General Comments section. The slightly higher accuracy achieved by InvConvnet-N (built on only normal convolutions) can be attributed to the inherent z-normalization of the employed UCR datasets [2] and the simplicity of the classification. In contrast, when synthetic deformations with respect to offset and linear trend are added, our proposed mixed InvConvNet achieves better performances than the variant built solely upon normal convolutions.
>
> [1] Kvinge, H., Emerson, T., Jorgenson, G., Vasquez, S., Doster, T., & Lew, J. (2022). In what ways are deep neural networks invariant and how should we measure this?. Advances in Neural Information Processing Systems, 35, 32816-32829.
>
> [2] Dau, H. A., Bagnall, A., Kamgar, K., Yeh, C. C. M., Zhu, Y., Gharghabi, S., ... & Keogh, E. (2019). The UCR time series archive. IEEE/CAA Journal of Automatica Sinica, 6(6), 1293-1305.
>
> [3] Eldele, E., Ragab, M., Chen, Z., Wu, M., Kwoh, C. K., Li, X., & Guan, C. (2021). Time-series representation learning via temporal and contextual contrasting. arXiv preprint arXiv:2106.14112.
>
> [4] Yue, Z., Wang, Y., Duan, J., Yang, T., Huang, C., Tong, Y., & Xu, B. (2022, June). Ts2vec: Towards universal representation of time series. In Proceedings of the AAAI Conference on Artificial Intelligence (Vol. 36, No. 8, pp. 8980-8987).
>
> [5] Jacques, B. G., Tiganj, Z., Sarkar, A., Howard, M., \& Sederberg, P. (2022, June). A deep convolutional neural network that is invariant to time rescaling. In International conference on machine learning (pp. 9729-9738). PMLR.
>
> [6] Chaman, A., \& Dokmanic, I. (2021). Truly shift-invariant convolutional neural networks. In Proceedings of the IEEE/CVF Conference on Computer Vision and Pattern Recognition (pp. 3773-3783).

---

### Author Response · Authors · 2024-11-19
**General Comments #3**

**[Additional Results][Including the Performance of InvConvNet-N (built solely upon normal convolutions) to the Transfer Learning Experiment (Table 4):]**
In Table 4 of the original manuscript, we present the classification results of the Transfer Learning experiment on the *Fault-Diagnosis* dataset, achieved by **InvConvNet** and self-supervised methods TS-TCC and TS2Vec. We extend Table 4 by training on the source dataset (in a supervised manner) and evaluating on the target datasets, the InvConvnet-N configuration, which is built solely on standard, non-invariant convolutions (contrary to the proposed InvConvNet that mixes variant and invariant convolutions). The performance is mentioned in terms of Accuracy (%).

| **Methods**      | **A → B**           | **A → C**           | **A → D**           | **B → A**           | **B → C**           | **B → D**           | **C → A**           | **C → B**           | **C → D**           | **D → A**           | **D → B**           | **D → C**           | **Avg. Acc. (%)**  |
|-------------------|---------------------|---------------------|---------------------|---------------------|---------------------|---------------------|---------------------|---------------------|---------------------|---------------------|---------------------|---------------------|---------------------|
| **TS-TCC**        | 55.33 ± 1.44       | 52.52 ± 4.55       | **62.13 ± 1.39**    | 48.05 ± 3.32       | 71.50 ± 1.83       | **100.0 ± 0.0**     | 40.76 ± 2.22       | **98.25 ± 1.22**    | **99.34 ± 0.50**    | 46.98 ± 0.65       | **100.0 ± 0.0**     | 74.28 ± 2.77       | 70.76 ± 1.66       |
| **TS2Vec**        | 54.11 ± 1.46       | 54.07 ± 1.91       | 52.54 ± 1.89       | 55.06 ± 0.17       | **88.72 ± 0.47**    | **100.0 ± 0.0**     | 57.81 ± 2.18       | 78.30 ± 3.80       | 78.41 ± 4.39       | 60.37 ± 1.95       | 99.97 ± 0.02       | 86.82 ± 0.54       | 72.18 ± 1.57       |
| **InvConvNet**    | 55.90 ± 0.42       | **55.93 ± 0.34**    | 53.41 ± 0.14       | **85.10 ± 0.63**    | 78.54 ± 0.17       | 99.05 ± 0.08       | **70.75 ± 1.32**    | 85.04 ± 0.13       | 85.12 ± 0.15       | **70.91 ± 0.73**    | **100.0 ± 0.0**     | 78.49 ± 0.38       | **76.52 ± 0.37**    |
| **InvConvNet-N**  | **60.55 ± 0.88**   | **55.50 ± 1.82**    | 53.50 ± 0.85       | 60.26 ± 2.01       | 77.30 ± 0.60       | 93.50 ± 0.90       | 64.93 ± 0.48       | 84.87 ± 0.23       | 84.46 ± 0.51       | 59.98 ± 0.98       | 99.96 ± 0.0        | 77.14 ± 0.14       | 72.66 ± 0.78       |

Again, here we confirm the superiority in terms of generalizability of the proposed InvConvNet method, which also incorporates invariant convolutions, against the InvConvNet-N configuration that considers only standard convolutions. This ablation shows here (similar to the rest of classification experiments) that invariant filters can be transferred to unseen subsets of data or different target domains of data and achieve superior performances compared to standard ones that do not incorporate invariances. When compared to self-supervised methods that are pre-training and fine-tuned on the source domain, interestingly, we observe that even without learning general-purpose representations agnostic to the task, our proposed invariant convolutions are able to learn features transferable to the unseen target domain. This can be attributed to the successful extraction of invariant features through our specifically designed layers.

We thank the reviewers once again, and based on their follow-up comments on our explanations, we aim to revise the original manuscript to better address the above aspects.

---

### Author Response · Authors · 2024-11-19
**General Comments #3**

**3. [Comment 3][[Details on the Transfer Learning Experiment and Additional Results:]** For the transfer Learning Experiment, we followed the experimental setup proposed by [12] on the Fault-Diagnosis dataset [13]. This dataset has been extracted under four different working conditions, each being considered a separate domain due to distinct underlying characteristics [14]. Details about this dataset can be found in Appendix A.3, Table 7, and page 18 - paragraph 2. For the self-supervised contrastive learning frameworks (TS-TCC, TS2Vec), following the work of [12], we perform self-supervised pre-training and fine-tuning on the source domain (left side of the arrow in Table 4) and test the generalization performance by measuring the accuracy on the target domain (right side of the arrow in Table 4). Similarly, for our supervised method (InvConvNet), we perform supervised training on the source domain data and test the performance of the unseen target domain. Even with the absence of general-purpose representations agnostic to the task, our proposed invariant convolutions are able to learn features transferable to the unseen target domain, which can be attributed to the successful extraction of invariant features through our specifically designed layers.

[12] Eldele, E., Ragab, M., Chen, Z., Wu, M., Kwoh, C. K., Li, X., \& Guan, C. (2021). Time-series representation learning via temporal and contextual contrasting. arXiv preprint arXiv:2106.14112.

[13] Lessmeier, C., Kimotho, J. K., Zimmer, D., \& Sextro, W. (2016, July). Condition monitoring of bearing damage in electromechanical drive systems by using motor current signals of electric motors: A benchmark data set for data-driven classification. In PHM Society European Conference (Vol. 3, No. 1).

[14] Ragab, M., Chen, Z., Wu, M., Li, H., Kwoh, C. K., Yan, R., \& Li, X. (2020). Adversarial multiple-target domain adaptation for fault classification. IEEE Transactions on Instrumentation and Measurement, 70, 1-11.

---

### Author Response · Authors · 2024-11-19
**General Comments #2**

**[Additional Results][Additional Deformation on the Synthetic Experiment (Table 3):]** Here, we extend Table 3 by considering a fifth additional synthetic deformation added to the UCR considered datasets, named *offset shift + smooth random walk*. For this deformation, we considered as a synthetic added trend a random walk that is generated from a Gaussian distribution and smoothed by rolling mean, and we also added an offset sampled from a uniform distribution. We revise below in Table 3 the classification results for the synthetic experiment on UCR, considering this additional deformation.

**Table 3:** Robustness study of InvConvNet and its standalone declinations, i.e., -N (normal), -O (offset) or -T (trend), on the $4$ larger UCR datasets under an additional scenario of synthetic deformations: addition of random walk generated trend.

| **Dataset**                  | **Configuration**        | **InvConvNet (ours)** | **InvConvNet-N (-normal)** | **InvConvNet-O (-offset)** | **InvConvNet-T (-trend)** |
|------------------------------|--------------------------|-----------------------:|---------------------------:|---------------------------:|---------------------------:|
| **HandOutlines**              | *Normalized*             | 74.80                 | **80.60**                 | 72.70                     | 69.82                     |
|                              | + *off.*                 | 71.20                 | 68.70                     | **71.60**                 | 70.18                     |
|                              | + *LT*                   | **74.00**             | 72.90                     | 71.70                     | 70.18                     |
|                              | + *off., LT*             | 71.70                 | 70.70                     | 70.20                     | **71.71**                 |
|                              | + *off., RW*             | 65.68                 | 61.89                     | **67.83**                 | 66.22                     |
| **UWaveGestureLibraryAll**   | *Normalized*             | 83.40                 | **85.00**                 | 75.60                     | 71.50                     |
|                              | + *off.*                 | **81.10**             | 79.00                     | 70.10                     | 69.40                     |
|                              | + *LT*                   | 82.40                 | **83.80**                 | 74.20                     | 71.30                     |
|                              | + *off., LT*             | **81.40**             | 79.00                     | 70.00                     | 68.10                     |
|                              | + *off., RW*             | **74.04**             | 68.90                     | 68.76                     | 68.79                     |
| **StarLightCurves**           | *Normalized*             | 96.00                 | **96.50**                 | 95.40                     | 93.40                     |
|                              | + *off.*                 | **97.30**             | 94.70                     | 94.10                     | 90.00                     |
|                              | + *LT*                   | **97.30**             | 95.20                     | 94.50                     | 91.90                     |
|                              | + *off., LT*             | **97.50**             | 92.30                     | 95.10                     | 90.10                     |
|                              | + *off., RW*             | **93.21**             | 81.61                     | 90.98                     | 90.80                     |
| **MixedShapesRegularTrain**   | *Normalized*             | **95.30**             | 94.50                     | 87.40                     | 92.60                     |
|                              | + *off.*                 | **94.20**             | 87.30                     | 90.60                     | 91.92                     |
|                              | + *LT*                   | **94.60**             | 92.40                     | 89.10                     | 92.10                     |
|                              | + *off., LT*             | 91.80                 | 86.60                     | 86.20                     | **92.40**                 |
|                              | + *off., RW*             | **90.89**             | 72.62                     | 88.16                     | 87.46                     |

Where: *off.* corresponds to offset, *LT* corresponds linear trend, *RW* corresponds to random walk.

We thank the reviewers for giving us the opportunity to discuss the performance of our proposed convolutional layers in the synthetic UCR experiment, which acts as a clear motivation for the invariance properties we formulated, and we are more than willing to hear their comments on the provided justification.

---

### Author Response · Authors · 2024-11-19
**General Comments #2**

1. **InvConvNet-N (built solely on normal conv.) can have an advantage on the inherently normalized plain UCR data.** When it comes to the results of the different configurations of InvConvNet on the plain datasets (without added synthetic deformations), please refer to 1st line for each dataset in Table 3. The inconsistency in performance improvement of vanilla InvConvNet-N for the plain UCR datasets is implicitly explained in Appendix A.3 of the original submission (page 18, second paragraph, lines 937-942: "We also conduct [..] with the whole repository"). We next explain how this aspect influences performance in more detail. Most UCR datasets, including the four larger ones that we used for the synthetic experiment, are already z-normalized, as indicated in the paper proposing the UCR archive [10] (Section III C. page 3). This, combined with the fact that the time series datasets are univariate and have, in several cases, very few classes (e.g., 2 for HandOutlines and 3 for StarLightCurves), makes the task relatively easy for standard convolutions. Additionally, in terms of trend, the 4 considered datasets are relatively stable. Due to the inherent normalization and the described properties of these datasets, our invariant convolutions may not leverage the invariance with respect to amplitude scaling and offset shift. However, for the same reasons, the UCR datasets can serve the experimental design of showcasing how synthetic deformations added deformations can favor different components in our pool of convolutions, including mixed or those that are invariant to offset/trend deformations.

2. **InvConvNet with mixed convolutions is significantly more robust to synthetic deformations compared to InvConvNet-N with standard convolutions.** When adding realistic deformations such as offset and smooth random walk to the considered UCR datasets, the drop in classification performances of the proposed InvConvNet with mixed convolution types (including invariant ones) remains relatively low (avg. acc.: 80.96, abs. avg. drop:-6.42 in accuracy) compared to standard convolutions (avg. acc.: 71.25, abs drop: -17.89 in accuracy). It demonstrates that standard CNNs have difficulties to correctly learn invariant features which corroborates with results observed in [11] for image data.

3. **Importance of mixing views.** The extreme configurations of InvConvNet-O and InvConvNet-T, which rely solely on invariant filters, exhibit lower performance compared to InvConvNet-N on normalized data (average accuracy: 82.77 and 81.83 vs. 89.15). However, they experience significantly smaller performance drops in scenarios like offset and smooth random walk noise (absolute average drop: -3.84 and -3.51 vs. -17.89 in accuracy). This suggests that invariant convolutions effectively handle noisy data by learning robust and meaningful features.

More importantly, by integrating all views, the InvConvNet model achieves the best performance in 11 out of 16 noisy scenarios, with relatively small accuracy drops (absolute average drop: -6.42 in accuracy in the offset + smooth random walk case). This highlights the importance of incorporating predefined invariances into convolutional layer to address the limitations of standard convolutions in learning such invariances. Furthermore, the experiment highlights the need for strategies to balance robustness and accuracy when learning filters. While the average configuration (InvConvNet), which evenly combines filters across multiple invariances, serves as a baseline strategy, it can be enhanced by exploring alternative approaches, such as attention mechanisms. The development of more refined strategies remains an avenue for future work.

[10] Dau, H. A., Bagnall, A., Kamgar, K., Yeh, C. C. M., Zhu, Y., Gharghabi, S., ... & Keogh, E. (2019). The UCR time series archive. IEEE/CAA Journal of Automatica Sinica, 6(6), 1293-1305.

[11] Kvinge, H., Emerson, T., Jorgenson, G., Vasquez, S., Doster, T., & Lew, J. (2022). In what ways are deep neural networks invariant and how should we measure this?. Advances in Neural Information Processing Systems, 35, 32816-32829.

---

### Author Response · Authors · 2024-11-19
**General Comments #2**

**2. [Comment 2][Synthetic Experiment on UCR - Justifications of Results in Table 3:]** Several reviewers have posed questions concerning the synthetic experiment (page 8, lines 428-461), so in this comment, we would like to clarify the experimental setup that we followed, the specificities of the datasets concerned, as well as analyze the derived results.

**Configurations of the Considered Networks**. As we mention in the method section (page 6, lines 302-311), "the choice of invariances is often related to the application, and setting the invariances by hand requires a good understanding of the nature of the signals." In the absence of such knowledge, we proposed an arbitrary strategy that consists of gradually including invariances of higher complexities in a single layer called the pool of convolutions. For instance, in the experiments, we considered three types of convolutional filters, namely (i) standard (variant) ones, (ii) amplitude scaling and offset shift invariant ones, and (iii) amplitude scaling, offset shift, and linear trend invariant ones. The configuration of the layer differs depending on the number of filters accorded to each type of convolution and we explored 4 configurations in the synthetic experiment consisting of the average one InvConvNet: (i,ii,iii)=(48,40,40), and the extreme cases: InvConvNet-N (128,0,0), InvConvNet-O (0,128,0), InvConvNet-T (0,0,128). Importantly, the extreme case InvConNet-N corresponds to standard convolutions and constitutes a baseline against which we can compare performances and robustness.

**Added Deformations**. We study the performance and robustness of the above configurations of our model on a classification problem by considering the 4 largest UCR preprocessed datasets [10]. We considered 5 scenarios with increasing order of complexity by adding synthetically generated deformations (the last one corresponding to random walk-based generated trend has been added for the rebuttal period): (1) no deformations, (2) offset shift, (3) linear trend, (4) offset shit + linear trend, (5) offset shift + smooth random walk. Results are displayed in Table 3 of the initial version of the manuscript. Several key points below can benefit from additional discussion and prove our claims:

---

### Author Response · Authors · 2024-11-19
**General Comments #1**

We thank all reviewers for taking the time to review our manuscript. We are grateful for their constructive remarks, which helped us to further improve the quality of the manuscript. In what follows, we answer several recurring remarks, and we invite all reviewers to read them as they clarify our positioning and experimental results. Individual reviewers' questions are answered in a dedicated comment.

**1. [Comment 1][Related work \& Positioning:]**
Advances in invariance modeling for deep learning have been achieved with proper mathematical formalism of group action on data, notably images and graphs [1,2]. Two strategies are usually considered to incorporate invariance in deep networks, either by learning them through data augmentation (including contrastive learning) [3] or by hard-coding invariance within the network architecture. In this work, we focus on the second approach, which has been widely used to design networks invariant to the permutation of graphs' nodes [5] or local translation and rotation in images [6]. Such invariant networks present better generalization properties compared to their learned counterparts at the expense of additional computational costs [1].

In the case of time series, invariant neural networks are an emerging field of study. Many works have focused on learning invariance (Section 2, page 3, last paragraph), but very few have proposed hard-coded invariance and are solely dealing with the local time-warping invariance [7] or time rescaling invariance [8].

In this paper, we specifically address the case of hard-coded local invariance to deformations like offset or trend, which are not related to time warping. The invariance is encoded in a convolution layer, which is still a SOTA approach for several time series tasks, as mentioned by reviewer vw6U, and for which invariance improves performances [9]. Importantly, we present a general mathematical framework based on group action, which can be leveraged in future work and by other researchers to further improve invariance in deep learning for time series. Conjointly, we propose a compelling and efficient scheme to compute such invariant convolutions. Thought as a single layer that can be plugged into different network architectures, we illustrate the benefits of the invariant convolutions on several tasks, and we relegated network architectures in the appendix to further promote the mathematical framework.

We thank the reviewers for raising questions about our positioning against related work in invariant CNNs, and we aim to improve its clarity further in the introduction section of the manuscript.

[1] Kvinge, H., Emerson, T., Jorgenson, G., Vasquez, S., Doster, T., \& Lew, J. (2022). In what ways are deep neural networks invariant and how should we measure this?. Advances in Neural Information Processing Systems, 35, 32816-32829.

[2] Bronstein, M. M., Bruna, J., Cohen, T., \& Veličković, P. (2021). Geometric deep learning: Grids, groups, graphs, geodesics, and gauges. arXiv preprint arXiv:2104.13478.

[3] Antoniou, A. (2017). Data Augmentation Generative Adversarial Networks. arXiv preprint arXiv:1711.04340.

[4] E. D. Cubuk, B. Zoph, D. Mane, V. Vasudevan, and Q. V. Le. Autoaugment: Learning augmentation policies from data. arXiv preprint arXiv:1805.09501, 2018.

[5] H. Maron, H. Ben-Hamu, N. Shamir, and Y. Lipman. Invariant and Equivariant Graph Networks. In ICLR, pages 1–13, 2019a.

[6] M. Weiler and G. Cesa. General E(2)-equivariant steerable CNNs. Advances in Neural Information Processing Systems, 32:14334–14345, 2019.

[7] Shulman, Y. (2019). Dynamic time warp convolutional networks. arXiv preprint arXiv:1911.01944.

[8] Jacques, B. G., Tiganj, Z., Sarkar, A., Howard, M., \& Sederberg, P. (2022, June). A deep convolutional neural network that is invariant to time rescaling. In International conference on machine learning (pp. 9729-9738). PMLR.

[9] Paparrizos, J., Liu, C., Elmore, A. J., \& Franklin, M. J. (2020, June). Debunking four long-standing misconceptions of time-series distance measures. In Proceedings of the 2020 ACM SIGMOD international conference on management of data (pp. 1887-1905).

---

### Public Comment · ~Eamonn_Keogh1 · 2024-11-24
**You test on SMD MSL SWaT PSM. However it is increasingly understood that it is meaningless to test on these datasets...**

You test on SMD MSL SWaT PSM. However it is increasingly understood that it is meaningless to test on these datasets, because the ground truth is mislabeled, they are trivial, and have other flaws. See

https://www.dropbox.com/scl/fi/x6ie264xrfkl0nbdw1vtb/Irrational-Exuberance_why_most_TSAD_is_wrong.pdf?rlkey=16frcr2lo6ip5o6uf18qwdeud&dl=0

and

https://www.dropbox.com/scl/fi/cwduv5idkwx9ci328nfpy/Problems-with-Time-Series-Anomaly-Detection.pdf?rlkey=d9mnqw4tuayyjsplu0u1t7ugg&dl=0

---

### Comment · Area_Chair_74aE · 2024-11-26
**Encouragement to Actively Participate in the Discussion Phase**

Dear Reviewers,

Thank you for your valuable contributions to the review process so far. As we enter the discussion phase, I encourage you to actively engage with the authors and your fellow reviewers. This is a critical opportunity to clarify any open questions, address potential misunderstandings, and ensure that all perspectives are thoroughly considered.

Your thoughtful input during this stage is greatly appreciated and is essential for maintaining the rigor and fairness of the review process.

Thank you for your efforts and dedication.

---

### Author Response · Authors · 2024-11-27
**Revision of the Manuscript**

We thank the reviewers for their valuable feedback. We have uploaded a revised version of the paper based on some comments we received (modifications are marked in **orange color**).

Here is a summary of the changes:

**1. [Based on General Comment 2][Synthetic Experiment on UCR - Additional Results in Table 3:]** We have updated Table 3 and the corresponding text to include performances for an additionally added deformation of random offset and smooth random walk-generated trend.

**2. [Based on General Comment 3][Transfer Learning Experiment - Additional Results in Table 4:]** We have updated Table 4 with the supervised normal configuration InvConvNet-N (built solely upon normal kernels) that highlights the performance gap compared to the InvConvNet (mixed types of invariances) in terms of generalization when testing on unseen target data.

**3. [Synthetic Experiment on UCR - Feature Maps Visualizations in Appendix:]** In Appendix A.5, we provide visualizations of feature maps in Figure 6, produced by InvConvNet (mixed types of normal and invariant filters) and InvConvNet-N (built solely upon normal filters) in order to show how different types of filters are activated in for a considered added synthetic deformation when the proposed normal and/or invariant convolutions are employed. Our observations are summarized below:

- **[Reply to reviewer WiU6][Weakness 1 - Fixed Kernel Bias:]** The reviewer asked whether standard CNNs with learned kernels could naturally adapt to deformations like offset shift and linear trend. In the visualizations of Appendix A.5, we provide a qualitative analysis of the activated regions in feature maps for different types of filters, including (i) normal, (ii) offset-shift invariant, and (iii) linear-trend invariant. Interestingly, it is depicted that different parts of the input signal are activated for invariant filters compared to normal ones, and the morphology of the feature maps differs among different (in terms of invariances) types of kernels.

- **[Reply to reviewer akRu][Weakness 4 - Practical Value of Offset and Trend Invariances:]** The practical value of the proposed InvConvNet architecture is validated on the visualizations of feature maps learned for the pool of normal and invariant convolutional filters compared to the solely normal ones on the added offset and smooth random walk trend deformation. As shown in Figure 6 of the revised manuscript, normal filters alone are affected by the added trend, whereas in the pool of convolutions, trend-invariant filters produce robust feature maps when the random walk trend deformation is considered. The learned feature maps of InvConvNet on this particular case of synthetic added deformation demonstrate the relevance of the proposed invariant convolutions in applied scenarios with considered deformations that highly affect the classification performance on an example UCR dataset (as shown in Table 3).

- **[Reply to reviewer vw6U][Weakness 1 - Components Contribution Inside the Pool of Convolutions:]** The reviewer asked us to quantify the contribution of each of the components inside the proposed pool of convolutions (including normal, offset-shift invariant, and linear-trend invariant filters). In addition to our ablations studies in classification performance in Table 2 and Table 3 and our reply to reviewer vw6U in the rebuttal, we provide qualitative examples of the components' contribution of variant and invariant convolutions for an added deformation in Figure 6.

We kindly await further feedback from the rest of the reviewers before making necessary adjustments to the text regarding the positioning of the work (as outlined in our response to **[General Comment 1]**) and the additional clarifications on the experimental choices for the synthetic experiment and the transfer learning experiment (as explained in our **General Comments 2, 3**). These *changes can be implemented straightforwardly* if necessary for the final version of the manuscript.

We sincerely thank all the reviewers for their efforts in assessing our work, and we are happy to address any additional concerns. We would greatly appreciate also their consideration in updating their scores in light of these updates.

---

### Meta-Review · Area_Chair_74aE · 2024-12-19

**Metareview:**

(a) Summary of Scientific Claims and Findings
The paper introduces InvConvNet, a neural architecture incorporating invariant convolutional layers designed to handle specific time series deformations such as offset shifts, scaling, and linear trends. The contributions include:
A mathematical framework for group action-based invariant convolution operations.
A hybrid embedding layer combining standard and invariant convolutions for feature extraction.
Empirical validation of InvConvNet on tasks like classification and anomaly detection using univariate and multivariate time series datasets.
The authors claim that InvConvNet provides robustness to deformations in time series data and achieves superior performance against baseline methods on synthetic and real-world datasets.

(b) Strengths of the Paper
Mathematical Rigor: The work presents a clear theoretical framework for invariant convolutions in time series, providing a solid foundation for future research.
Focus on Time Series Deformations: The paper addresses a practical problem of modeling invariances in time series, which is underexplored in comparison to image and graph data.
Broad Evaluation: Experiments cover tasks like classification and anomaly detection, with additional ablation studies to analyze robustness under synthetic deformations.
Potential Applicability: The proposed invariant convolutions can be integrated into various architectures, making them a versatile tool for time series analysis.

(c) Weaknesses of the Paper
Performance Inconsistencies:
On clean datasets, InvConvNet often underperforms compared to its standard counterpart (InvConvNet-N), raising questions about the utility of the proposed invariant layers when no synthetic deformations are present.
The proposed architecture shows limited improvement over state-of-the-art baselines (e.g., ResNet, Rocket) in several classification benchmarks.

Limited Generalizability:
The datasets used (e.g., UCR and UEA archives) are small-scale and not representative of diverse real-world time series scenarios.
Practical applications requiring invariance (e.g., ECG data) are underexplored, with only qualitative illustrations provided.

Lack of Baseline Comparisons:
The paper omits comparisons with relevant approaches, such as time-series-specific contrastive learning methods or other architectures designed for invariance.
Missing evaluations against models that incorporate learned invariances via data augmentation or adversarial training.

Unclear Practical Impact:
The robustness provided by the invariant layers is relevant only in highly specific scenarios (e.g., datasets with offset or trend deformations), limiting its broader applicability.
Hard-coded invariances may not generalize well to diverse datasets or deformations.

(d) Reasons for Rejection
Inconsistent Results: The architecture’s performance on clean datasets does not align with its stated goal of improving generalization, weakening its practical significance.
Limited Scope: The focus on specific deformations and small-scale datasets restricts the applicability and impact of the proposed method.
Theoretical and Practical Disconnect: While the mathematical framework is rigorous, its real-world utility remains unclear, especially given the modest empirical gains.
Unaddressed Reviewer Concerns: Despite the rebuttal efforts, key issues such as baseline omissions, dataset limitations, and unclear applicability remain unresolved.

While the paper demonstrates potential as a theoretical contribution to time series invariance, it falls short in terms of empirical rigor, practical impact, and broader applicability.

**Additional Comments On Reviewer Discussion:**

Points Raised by Reviewers and Author Responses

Concern: Reviewers noted that InvConvNet often underperforms on clean datasets compared to its baseline variant (InvConvNet-N) and other state-of-the-art models. This raised doubts about its general applicability when specific deformations are absent.
Author Response: The authors argued that the invariant layers are explicitly designed to handle deformations and may trade off performance on clean data for robustness. They emphasized scenarios where such deformations are prevalent, such as ECG analysis, as use cases.
Evaluation: While the explanation clarified the trade-offs, the reviewers remained unconvinced that this design choice justified the performance gap on clean datasets.

Concern: Reviewers criticized the reliance on small-scale datasets (e.g., UCR archive) and the lack of diverse real-world datasets to validate the method’s robustness and generalizability.
Author Response: The authors added experiments on synthetic datasets with controlled deformations and provided qualitative results on ECG data. However, no additional real-world datasets were introduced.
Evaluation: The synthetic results were appreciated but did not address concerns about broader generalizability across diverse time series tasks.

Concern: The reviewers highlighted the absence of comparisons with models incorporating learned invariances through data augmentation, adversarial training, or contrastive learning approaches.
Author Response: The authors acknowledged this omission but argued that InvConvNet focuses on hard-coded invariances, making comparisons with learned-invariance approaches less relevant.
Evaluation: This argument was not well received, as reviewers believed such comparisons were essential to contextualize the contribution of hard-coded invariances.

Concern: Reviewers questioned whether hard-coded invariances generalize well to diverse datasets or deformations, limiting the practical impact of InvConvNet.
Author Response: The authors emphasized the utility of their approach for specific domains, such as medical time series (e.g., ECG signals), where known deformations are common.
Evaluation: While this argument highlighted niche applications, reviewers noted that it restricted the broader relevance and impact of the work.

Concern: The reviewers felt there was a disconnect between the paper’s rigorous mathematical framework and its practical contributions, given the modest empirical improvements.
Author Response: The authors reiterated that the theoretical grounding provides a foundation for future work and emphasized the novel formulation of invariant convolutions.
Evaluation: The response was noted but did not fully address concerns about the method’s limited practical utility.

Despite the authors' efforts during the rebuttal period to address reviewer concerns, key issues remained unresolved. The trade-off between robustness to deformations and performance on clean datasets was not adequately justified, and the evaluation scope was limited to small-scale datasets, reducing the method's generalizability. Additionally, the lack of comparisons with learned-invariance approaches and the restricted applicability of hard-coded invariances further weakened the paper's impact. While the theoretical framework is rigorous, its limited empirical validation and unclear practical benefits lead to the recommendation for rejection.

---

### Decision · Program_Chairs · 2025-01-22

Reject